# Evolution of DNA methylation in the human brain

Hyeonsoo Jeong [1,9], Isabel Mendizabal [1,2,9], Stefano Berto[3], Paramita Chatterjee [1], Thomas Layman[1], Noriyoshi Usui [3,4], Kazuya Toriumi [3,5], Connor Douglas [3], Devika Singh[1], Iksoo Huh [1,6], Todd M. Preuss[7,8], Genevieve Konopka [3✉] & Soojin V. Yi [1✉]

DNA methylation is a critical regulatory mechanism implicated in development, learning, memory, and disease in the human brain. Here we have elucidated DNA methylation changes during recent human brain evolution. We demonstrate dynamic evolutionary trajectories of DNA methylation in cell-type and cytosine-context specific manner. Specifically, DNA methylation in non-CG context, namely CH methylation, has increased (hypermethylation) in neuronal gene bodies during human brain evolution, contributing to human-specific down-regulation of genes and co-expression modules. The effects of CH hypermethylation is particularly pronounced in early development and neuronal subtypes. In contrast, DNA methylation in CG context shows pronounced reduction (hypomethylation) in human brains, notably in cis-regulatory regions, leading to upregulation of downstream genes. We show that the majority of differential CG methylation between neurons and oligodendrocytes originated before the divergence of hominoids and catarrhine monkeys, and harbors strong signal for genetic risk for schizophrenia. Remarkably, a substantial portion of differential CG methylation between neurons and oligodendrocytes emerged in the human lineage since the divergence from the chimpanzee lineage and carries significant genetic risk for schizophrenia. Therefore, recent epigenetic evolution of human cortex has shaped the cellular regulatory landscape and contributed to the increased vulnerability to neuropsychiatric diseases.

[1] School of Biological Sciences, Georgia Institute of Technology, Atlanta, GA, USA. [2] Center for Cooperative Research in Biosciences (CIC bioGUNE), Basque Research and Technology Alliance (BRTA), Bizkaia Technology Park, Derio, Spain. [3] Department of Neuroscience, UT Southwestern Medical Center, Dallas, TX, USA. [4] Center for Medical Research and Education and Department of Neuroscience and Cell Biology, Graduate School of Medicine, Osaka University, Suita, Osaka, Japan. [5] Schizophrenia Research Project, Department of Psychiatry and Behavioral Sciences, Tokyo Metropolitan Institute of Medical Science, Tokyo, Japan. [6] College of Nursing and The Research Institute of Nursing Science, Seoul National University, Seoul, South Korea. [7] Division of Neuropharmacology and Neurologic Diseases, Yerkes National Primate Research Center, Emory University, Atlanta, GA, USA. [8] Department of Pathology, Emory University School of Medicine, Atlanta, GA, USA. [9] These authors contributed equally: Hyeonsoo Jeong, Isabel Mendizabal. ✉email: Genevieve. Konopka@utsouthwestern.edu; soojinyi@gatech.edu

DNA methylation is a stable epigenetic modification of genomic DNA with critical roles in brain development[1–3]. To understand the contribution of DNA methylation to human brain-specific gene regulation and disease susceptibility, it is necessary to extend our knowledge of evolutionary changes in DNA methylation during human brain evolution. It was previously suggested that human brain-specific CG methylation may be associated with human brain-specific regulation of gene expression[4,5]. However, these studies used bulk tissues, while DNA methylation is known to vary substantially between cell types. Cell-type-specific epigenetic marks, including DNA methylation and histone modifications, are implicated in cell-type-specific gene expression and disease susceptibility in humans[6,7]. Data from bulk tissues can be biased toward specific cell types and consequently, underpowered to detect cell-type-specific evolutionary changes[8,9]. Therefore, to fully understand the role of DNA methylation in human brain evolution, it is necessary to study cell-type-specific changes of DNA methylation.

Moreover, DNA methylation at non-CG contexts (CH methylation, where H = A, C, T) is relatively abundant in brains, where it is associated with postnatal neuronal maturation and cell-type-specific transcriptional activity[1,10,11]. Despite such importance, the evolutionary trajectories and significance of CH methylation during human brain evolution remain little understood.

In this work, we present comparative analyses of neuron- and oligodendrocyte-specific whole-genome DNA methylomes of humans, chimpanzees, and rhesus macaques. We further integrated these data with transcriptome data from the same individuals[8] and recent data from studies of bulk and cell-type-specific epigenetic and transcriptomic modifications of human brains[2,12–14]. By doing so, we show that dramatic changes of DNA methylation have occurred in a cell-type and cytosine-context-specific manner during human brain evolution. These DNA methylation changes are deeply implicated in the human brain-specific regulatory landscape and disease susceptibility. Our work extends the knowledge of the unique roles of CG and CH methylation in human brain evolution and offers a new framework for investigating the role of the epigenome evolution in connecting the genome to brain development, function, and diseases.

## Results

**Distinctive methylomes of neurons and oligodendrocytes in human and non-human primate prefrontal cortex.** We generated cell-type-specific DNA methylomes of sorted nuclei from post-mortem brain samples of humans[7], chimpanzees (*Pan troglodytes*), and rhesus macaques (*Macaca mulatta*). We selected Brodmann area 46 (BA46) from dorsolateral prefrontal cortex (also referred to as "prefrontal cortex" or "cortex" henceforth), which is involved in higher-order cognitive functions that have likely undergone marked changes in human evolution[15,16]. Neuronal (NeuN+) and oligodendrocyte (OLIG2+) cell populations were isolated using fluorescence-activated nuclei sorting (FANS) as previously described[7,8]. We used whole-genome bisulfite sequencing (WGBS) to generate DNA methylomes at nucleotide resolution for NeuN+ and OLIG2+ populations (Supplementary Fig. 1). Altogether, we compared 25, 11, 15 NeuN+ methylomes and 20, 11, 13 OLIG2+ methylomes from human, chimpanzee, and rhesus macaque, respectively (Supplementary Data 1 and 2). We also performed whole-genome sequencing (WGS) of the same individuals (Supplementary Data 3). Polymorphic sites at cytosines (i.e., C to T for forward strand and G to A for reverse strand) were excluded to avoid spurious methylation calls due to the technical limitation of distinguishing bisulfite-converted thymine from unmethylated cytosine (Supplementary Data 4). The mean coverages for the WGBS and WGS data are 20.6X (±8.8) and 23.2X (±5.9), respectively.

We found that as in humans, non-human primate prefrontal cortex is highly methylated at CG sites, and NeuN+ DNA is more highly methylated than OLIG2+ DNA ($P < 10^{-10}$, two-sample Kolmogorov-Smirnov test, Fig. 1a). In comparison, CH methylation occurs in much lower frequencies than CG methylation, and is nearly exclusive to NeuN+ DNA in humans[1,7] and non-human primates (Fig. 1a). Interestingly, neurons of humans and chimpanzees have significantly more highly CH methylated sites than those of rhesus macaques and mice (Fig. 1a, $P = 4.3 \times 10^{-5}$, Kruskal–Wallis test), indicating that brain CH methylation may have increased in human and ape brains. In turn, human brains show greater CH methylation compared to chimpanzee brains (Fig. 1a, $P = 0.03$, Mann–Whitney $U$-test using proportions of mCH > 10%).

Principal component analyses demonstrate that cell-type explains the largest amount of variation in both methylation contexts, followed by species (Fig. 1b). Since OLIG2+ DNA is largely devoid of CH methylation, there is little separation of species for CH OLIG2+ (Fig. 1b). As the genomic patterns and cellular distributions of CG and CH methylation are highly distinct from each other (Fig. 1b), we analyzed them separately.

**Conservation and divergence of cell-type-specific CG methylation.** Owing to the high rate of CG mutations associated with DNA methylation[17], the rate of CG loss is significantly higher compared to those of CH (Supplementary Fig. 2). Consequently, only 9.6 million CG sites (out of 28 million total human CGs) are conserved in all three species (Supplementary Fig. 3) and these sites are biased toward hypomethylation (Supplementary Fig. 4). As expected, evolutionarily conserved CpG sites co-localize with CpG islands and exons[18,19] (Supplementary Fig. 5). In addition, evolutionarily conserved CpGs and human-specific CpGs are enriched in distinctive transcription factor binding motifs (Supplementary Fig. 6 and Supplementary Data 5). Interestingly, HOX and FOX transcription factor families, among others, are significantly more often associated with human-specific CpGs than conserved CpGs (Supplementary Fig. 6 and Supplementary Data 5).

To avoid bias associated with CG conservation, we first identified differentially methylated regions (DMRs) that distinguish humans and chimpanzees using conserved sites (21 out of 25 million CGs analyzed), and subsequently added DNA methylation data from rhesus macaques to polarize direction of evolutionary change (Methods). In this analysis, we applied methods developed for the analysis of whole-genome bisulfite sequencing data to identify species, cell-type and interaction effects on DNA methylation while taking into account variation due to sex, age, and bisulfite conversion rates (Methods).

Non-human primate methylomes of NeuN+ and OLIG2+ are highly distinct from each other and show clear clustering of cell types in each species (Supplementary Fig. 7), as in humans[7]. There are 56,532 CG DMRs (75.9 Mbp) between NeuN+ and OLIG2+ DNA that are conserved in all three species (Fig. 1c and Supplementary Data 6). These conserved DMRs account for nearly 50% of all DMRs between NeuN+ and OLIG2+ in humans (Fig. 1d). Consequently, a large portion of differential CG methylation between NeuN+ and OLIG2+ DNA originated before the divergence of hominoids and catarrhine monkeys. Enrichment tests utilizing cis-regulatory interactions based on long-range regulatory domains[20] show that these regions are highly enriched in genes harboring functions specific to neurons

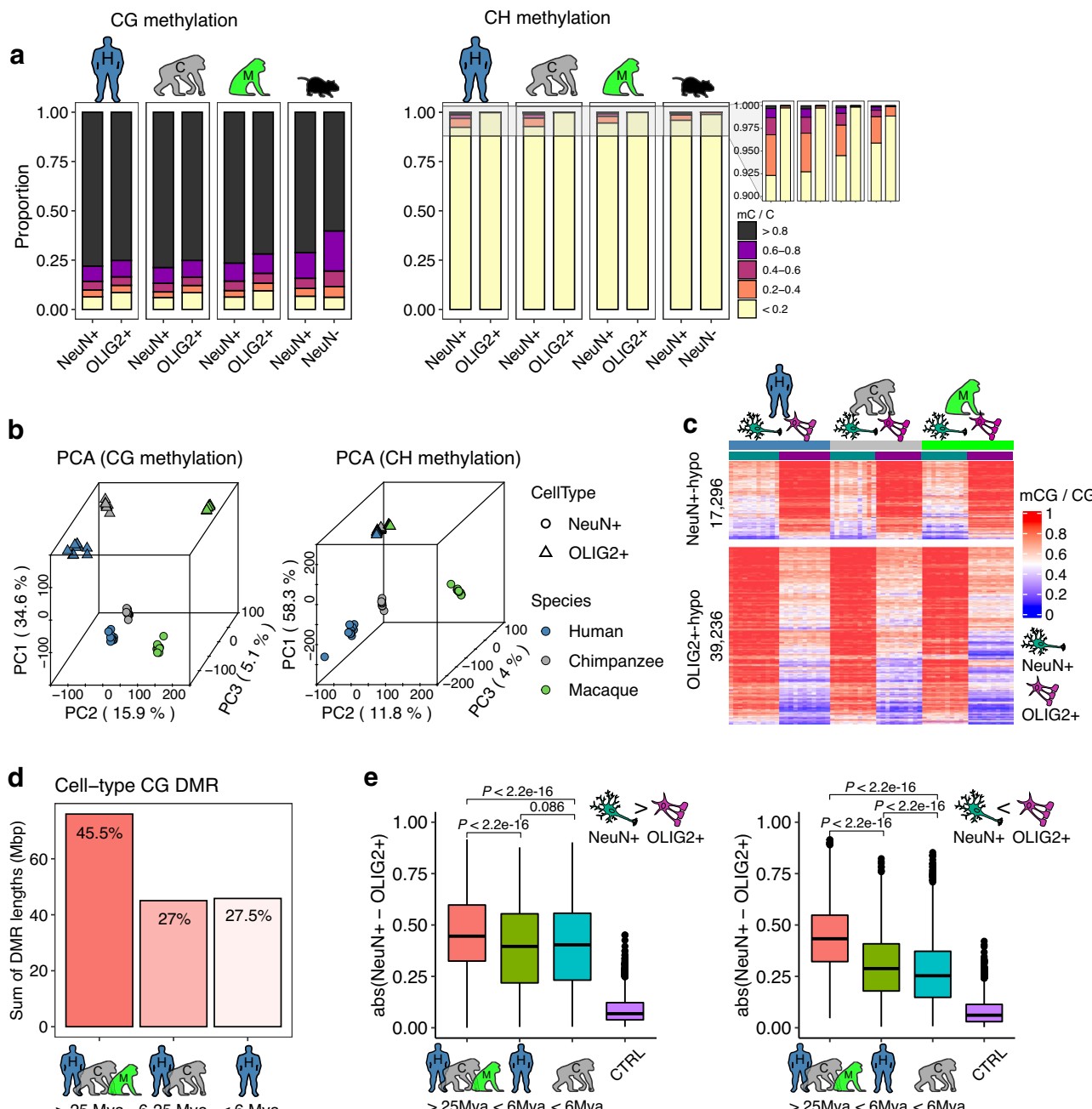

**Fig. 1 CG and CH methylation in neurons (NeuN+ cells) and oligodendrocytes (OLIG2+ cells) in human and non-human primate prefrontal cortex. a**
The proportions of methylated CG and CH sites. Human and non-human primate neurons and oligodendrocytes are highly CG methylated. Human and non-human primate neurons show low levels of CH methylation and oligodendrocytes show even lower levels. CH methylation is highest in human neurons, followed by chimpanzees, rhesus macaques, and mice. **b** Principal component analysis of methylated cytosines in two contexts (CG and CH). The top two principal components (PCs), PC1 and PC2, distinguish cell-type and species, respectively. **c** CG methylation levels in neurons (left columns for each species) and oligodendrocytes (right columns for each species). A greater number of DMRs are hypermethylated in neurons (red, in the left columns) compared to oligodendrocytes (right columns). **d** Approximately half (45.5%) of CG DMRs differentially methylated between NeuN+ and OLIG2+ cells are conserved in all three species, with 27% conserved between humans and chimpanzees, and 27.5% specific to the human. **e** The absolute methylation difference of NeuN+ and OLIG2+ cells is highest for DMRs conserved in all three species (39,202 and 17,284 DMRs hypermethylated in neurons and oligodendrocytes, respectively) compared to those specific to humans (3103 and 5361 DMRs hypermethylated in neurons and oligodendrocytes, respectively) or chimpanzees (4370 and 2989 DMRs hypermethylated in neurons and oligodendrocytes, respectively). DNA methylation differences between NeuN+ and OLIG2+ cells calculated from genomic regions serving as statistical control (CTRL), with a matched number of CG and G + C nucleotide contents, are also displayed. Statistical significance was computed using two-sided Mann–Whitney $U$-test. Box represents a range from the first quartile to the third quartile. The line in the box indicates the median value. The minima and maxima are within 1.5 times the distance between the first and third quartiles from box. Source data are provided as a Source Data file. Cell-type images were color modified from the original image, which was created by Akiyao and available at the Wikimedia Commons under the Creative Commons license (https://creativecommons.org/licenses/by-sa/3.0/deed.en).

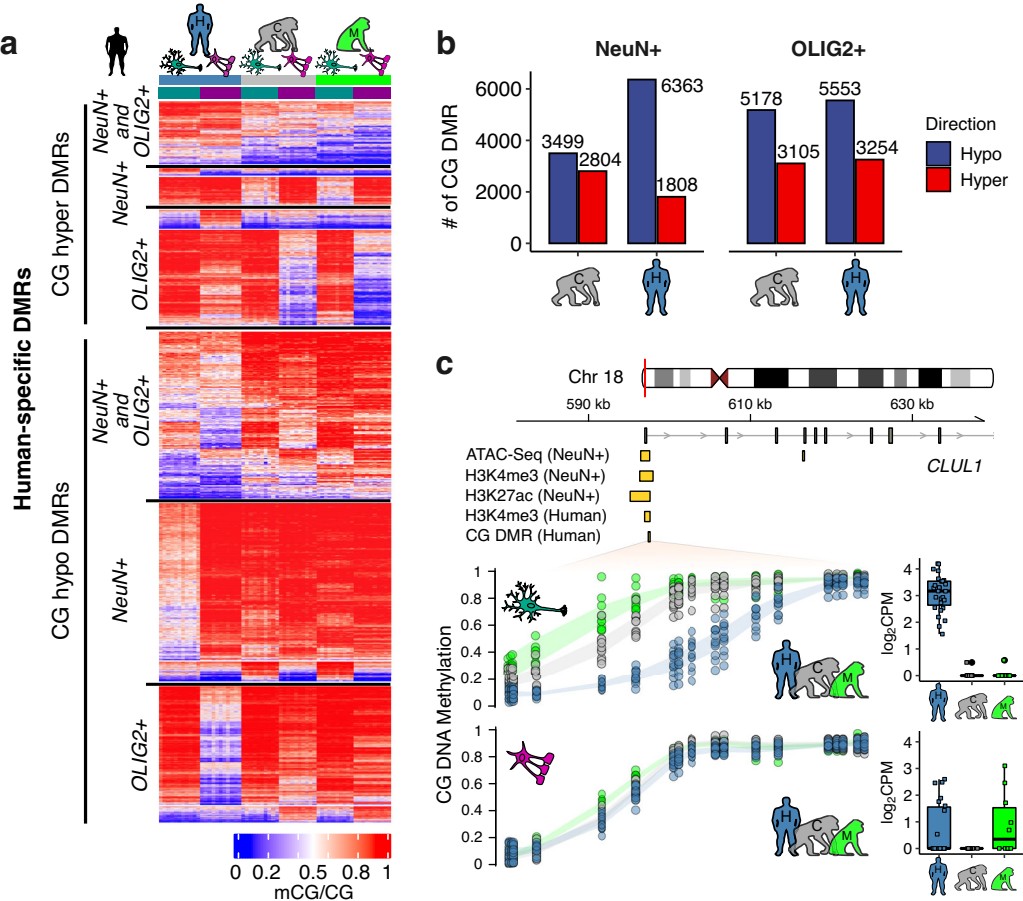

**Fig. 2 Evolutionary changes in CG methylation. a** Heatmap representation of mean DNA methylation of all 23,703 human DMRs in the three species illustrates dramatic reduction of CG methylation in human prefrontal cortex, especially in neurons. **b** Numbers of DMRs in NeuN+ and OLIG2+ cells in human and chimpanzee frontal cortex. **c** An example of the relationship between human neuron-hypo CG DMR and other epigenetic marks in the *CLUL1* locus, a gene widely expressed in the brain. This DMR overlaps with multiple other epigenetic marks of active chromatin in the human brain, including neuron-specific ATAC-Seq peak, neuron-specific H3K4me3 peak, neuron-specific H3K27ac peak. This DMR also overlaps with a human-specific brain H3K4me3 peak compared to chimpanzee and macaque. Box represents a range from the first quartile to the third quartile. The line in the box indicates the median value. The minima and maxima are within 1.5 times the distance between the first and third quartiles from box. Source data are provided as a Source Data file. Cell-type images were color modified from the original image, which was created by Akiyao and available at the Wikimedia Commons under the Creative Commons license (https://creativecommons.org/licenses/by-sa/3.0/deed.en).

and oligodendrocytes (Supplementary Data 7). For example, we show one conserved DMR spanning the whole *QKI* locus (Supplementary Fig. 7c). This gene, which is an RNA binding protein involved in myelination and oligodendrocyte differentiation[21], is covered entirely by a DMR in all three species so that it is hypomethylated in oligodendrocytes while hypermethylated in neurons. Gene expression data from matched samples[8] shows that *QKI* is significantly upregulated in oligodendrocytes compared to neurons in all three species ($P < 10^{-7}$ in all three species, Methods). This example illustrates that differential DNA methylation may facilitate cell-type-specific regulation in human and non-human primate brains. Interestingly, the absolute methylation difference between neurons and oligodendrocytes was significantly more pronounced in the evolutionarily "old" DMRs conserved in all three species compared to those recently evolved in human (Fig. 1e).

**Pronounced CG hypomethylation of human prefrontal cortex and human neuron-specific regulatory landscape.** We found 23,703 CG DMRs (13.1 Mbp) that experienced differential CG methylation since the divergence of humans and chimpanzees

(Methods, Fig. 2a, b), distributed across different functional categories, including regions currently annotated as non-coding intergenic (Supplementary Fig. 8). These CG DMRs include 7861 for which both cell types are differentially methylated between humans and chimpanzees (4253 human-specific and 3608 chimpanzee-specific CG DMRs, based on the comparison to macaques). The rest of the CG DMRs show DNA methylation changes in a cell-type-specific manner in each species (Fig. 2a and Supplementary Data 8). Interestingly, CG DMRs were found more often than expected near previously identified brain mQTLs[22] (Supplementary Fig. 9), suggesting that some genomic regions might be more susceptible to genetic changes that affect DNA methylation. This is in line with the observation that the evolution of DNA methylation is associated with underlying genetic sequences[23].

To provide insights into how DNA methylation changes at cell-type level have affected gene expression and other functional features, in the following we present results of DNA methylation analyses for each cell-type, combining DMRs that are common in both cell types and DMRs that are cell-type-specific in each species (Methods). While most previous studies focused on neurons, recent studies have begun to unveil the functional and

evolutionary importance of oligodendrocytes-specific changes[8,24]. Indeed, we identified a substantial number of human-derived hypomethylated DMRs specific to oligodendrocytes (Fig. 2a, b).

CG DMRs tend to show reduction of DNA methylation (hypomethylation) in human prefrontal cortex compared to chimpanzee in both cell-types (Fig. 2b). Their enrichments in promoters and the 5' end of genes (Supplementary Fig. 8) suggest impacts on gene regulation, as hypomethylation near transcription start sites is significantly associated with upregulation of gene expression[25]. Indeed, genes harboring human-hypomethylated CG DMRs are significantly enriched in human upregulated genes compared to chimpanzees, in the same oligodendrocyte and neuron cell populations[8] (Supplementary Data 9). These results indicate widespread and significant contributions of recent CG hypomethylation to the transcriptional landscape of the human brain.

Human neurons in particular harbor a large number of hypomethylated CG DMRs compared to chimpanzee neurons (Fig. 2b, 6363 hypomethylated CG DMRs in human neurons versus 3499 hypomethylated DMRs in chimpanzee neurons, OR (odds ratio) $= 2.82$, $P = 5.5 \times 10^{-20}$, chi-square test). Taking advantage of recent functional genomics data from human neurons, we show that human neuron-specific hypomethylated CG DMRs (referred to as "neuron-hypo CG DMRs" henceforth) mark active regulatory regions of the neuronal genome (Supplementary Fig. 10). Specifically, a substantial portion of human neuron-hypo CG DMRs co-localize with brain-specific enhancers (Supplementary Fig. 11), as well as other recently characterized cell-type-specific human brain epigenetic marks, including neuron-specific H3K27ac (fold-enrichment $= 3.1$, $P < 0.01$, permutation test), H3K4me3 (fold-enrichment $= 8.5$, $P < 0.01$), and ATAC-Seq (fold-enrichment $= 8.2$, $P < 0.01$) peaks[6,26] (Supplementary Fig. 10). For example, we show a human-specific neuron-hypo CG DMR in a 5' region of the *CLUL1* locus, which overlaps with other epigenomic signatures of active chromatin marks observed in human neurons (Fig. 2c). Even though its functional role is not resolved yet, previous studies showed that this gene is highly expressed across different brain regions[27]. Using matched gene expression data, we show that this locus is upregulated in a cell-type and lineage-specific manner in human neurons (Fig. 2c), consistent with the role of human-specific neuron CG hypomethylation.

In order to reveal the target genes of these epigenetically coordinated regulatory elements in human neurons, we integrated three-dimensional maps of chromatin contacts from the developing human cortex[28]. This analysis identified 213 enhancer-promoter pairs (Supplementary Fig. 12a, fold-enrichment $= 2.45$, $P < 0.01$, permutation test), supporting physical chromatin interactions between spatially adjacent human neuron-hypo CG DMRs in human neuron nuclei (Supplementary Data 10). Interestingly, genes affected by these enhancer-promoter interactions are enriched in functional categories, including neuron differentiation and development (Supplementary Data 11).

We also explored the co-occurrence of epigenetically identified regulatory elements with those emerging from DNA sequence analyses. Human hypomethylated CG DMRs, while enriched for both conserved and human-specific CpGs, are significantly associated with binding motifs for three transcription factors, including two Forkhead box factors (FOXP1 and FOXK1) and the nuclear factor 1 C-type, NFIC (Supplementary Data 12). The presence of these motifs further associates with greater hypomethylation of the DMRs themselves, as well as with increased expression of downstream genes (Supplementary Fig. 13). Furthermore, non-coding human accelerated regions (ncHAR) significantly overlap with human-specific hypomethylated CG

DMRs (Supplementary Fig. 12a, fold-enrichment $= 4.45$, $P < 0.01$, permutation test). In contrast, chimpanzee-specific hypomethylated CG DMRs did not show significant patterns (Supplementary Fig. 12b). Notably, ncHARs also show an excess of three-dimensional interactions with distant human hypo CG DMRs, which include seven experimentally validated human brain enhancer ncHARs[29]. In addition, human neuron-hypo CG DMRs frequently co-occur with human neuron-specific histone H3-trimethyl-lysine 4 (H3K4me3) modification[13] (fold-enrichment $= 18.1$, empirical $P$-value $< 0.01$, permutation test). Taken together, these results demonstrate the confluence of human-derived genetic and epigenetic innovations, and that CG hypomethylation of human neurons contributed to the active chromatin landscape of human prefrontal cortex in a cell-type-specific manner.

**Signature of evolutionarily recent CH hypermethylation in human neurons.** CH methylation is limited to a few cell types in the body[30,31], and occurs at much lower frequency than CG methylation (Fig. 1a). Nucleotide substitution rates at CH sites and CH methylation do not have a significant correlation[32]. Consequently, we were able to follow the evolutionary dynamics of CH methylation for the majority of CH positions. Among the 1.1 billion CH positions examined in the human genome, 716 million sites (71.2%) were found in the three species we examined (Methods). We found 51.9 million CH sites hypermethylated in NeuN+ compared to OLIG2+ DNA (FDR $< 0.05$). Among these, 23.6 million sites (45.5%) show NeuN+ DNA hypermethylation in all three species. Human and chimpanzee neurons share an additional 16.3 million (31.4%) CH hypermethylated sites not found in macaque (Fig. 3a). Moreover, an additional 3.1 million CH sites gained methylation in the human neurons (Supplementary Fig. 14), which is a significant excess compared to the 2.2 million sites gained via CH methylation in the chimpanzee neurons (OR $= 1.54$, 95% CI 1.534–1.546, $P < 10^{-20}$, chi-square test). Thus, in contrast to the pronounced hypomethylation in the CG context, human neurons are predominantly hypermethylated (Fig. 3b) compared to other primates.

CH methylation of gene bodies is one of the strongest predictors of repression of gene expression in humans and mice[1,7,10,33]. We find similarly strong repressive effects of genic CH methylation on gene expression in human and non-human primate neurons (Supplementary Fig. 15). Moreover, differential CH methylation between species is strongly negatively correlated with gene-expression differences between species, indicating that the change of CH methylation is a major determinant of neuronal transcriptional divergence (Fig. 3c).

**Distinctive evolutionary signatures of CG and CH methylation on the human neuronal transcriptome.** We have demonstrated that DNA methylation at different cytosine contexts shows distinctive patterns during the recent evolutionary history of human brains. Specifically, the pronounced hypomethylation in CG context, associated with active cis-regulatory elements, contrasts with the repressive hypermethylation observed at CH sites in gene-bodies in human neurons. Given that both types of methylation correlate with gene expression[1,10,25,34] (Supplementary Fig. 16), we analyzed their effects jointly using tools designed to measure independent effects of highly correlated variables[35]. These analyses point to significant and independent effects of both CG hypomethylation and CH hypermethylation (Supplementary Data 13 and 14). Compared to chimpanzees, genes upregulated in human neurons are more likely to have been impacted by CG hypomethylation at promoters, while those downregulated are prone to genic CH hypermethylation

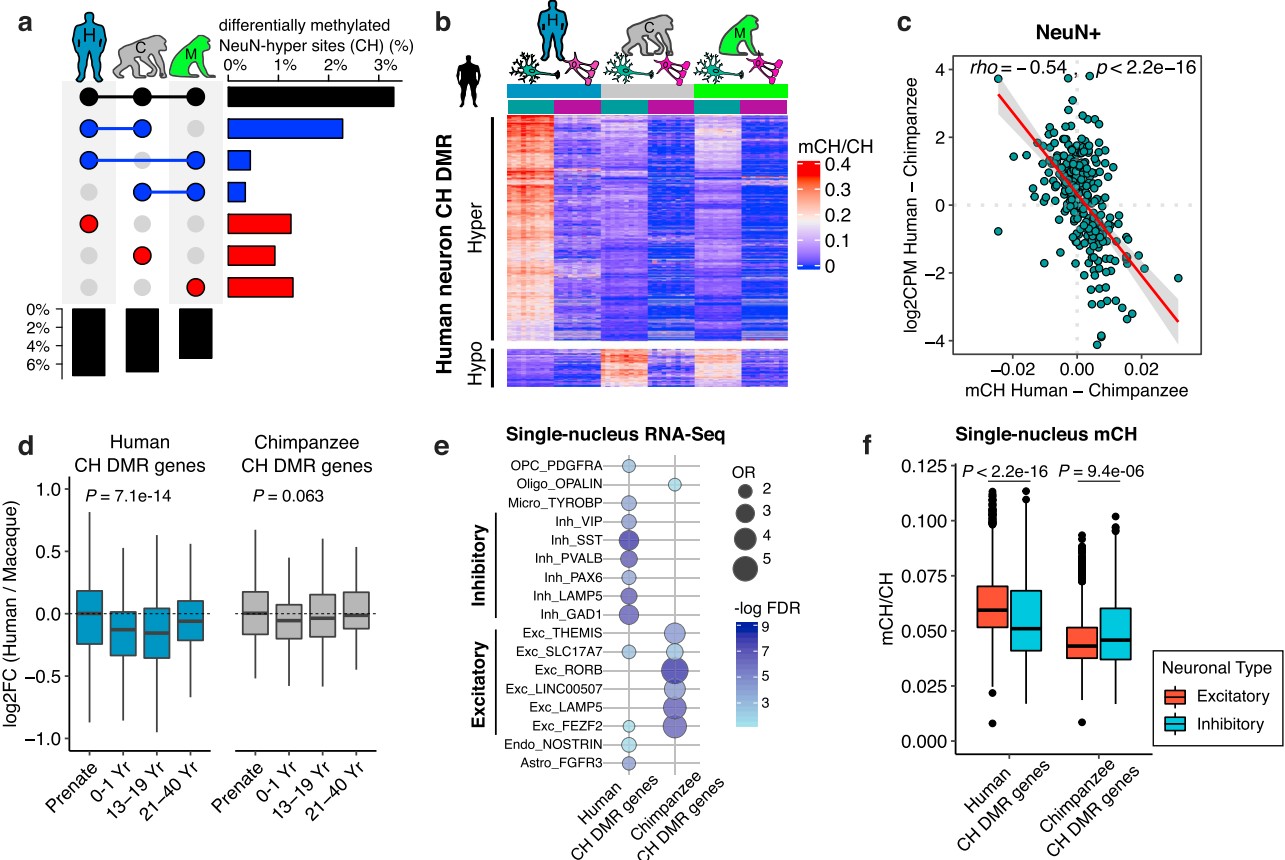

**Fig. 3 CH hypermethylation is significantly higher in human neurons compared to other primates. a** Differences in the proportions of sites with neuronal CH methylation between species. **b** Mean methylation levels of human-specific CH DMRs demonstrate pronounced hypermethylation of human neurons. **c** CH methylation between humans and chimpanzees strongly predicts gene expression difference. The shaded band represents the 95% confidence interval for the fitted regression line. **d** Gene expression fold-change between human and macaque in CH DMR genes across developmental time points (Human CH DMR genes, $n = 450$ and Chimpanzee CH DMR genes, $n = 144$). Macaque samples were age-matched to human developmental time points in a previous study[14]. Statistical significance was computed using Kruskal–Wallis test (two-sided). **e** Enrichment of human and chimpanzee CH DMR genes in specific cell-types. Human CH DMR genes are enriched in inhibitory neurons, whereas chimpanzee CH DMR genes are enriched in excitatory neurons. In each gene set, genes expressed in at least 50% of the cells that are statistically significant (FDR < 0.05 and $\log_2$FC > 0.3) are included. Cell-type data are from human medial temporal gyrus (MTG)[36]. **f** CH methylation of neuronal subtypes for CH DMR genes using methylation of single nuclei from the human frontal cortex[12]. Human CH DMR genes are hypomethylated in inhibitory neurons, whereas chimpanzee CH DMR genes are hypomethylated in excitatory neurons (excitatory neurons, $n = 1879$ and inhibitory neurons, $n = 861$). Statistical significance was computed using two-sided Mann–Whitney $U$-test. Box represents a range from the first quartile to the third quartile. The line in the box indicates the median value. The minima and maxima are within 1.5 times the distance between the first and third quartiles from box. Source data are provided as a Source Data file. Cell-type images were color modified from the original image, which was created by Akiyao and available at the Wikimedia Commons under the Creative Commons license (https://creativecommons.org/licenses/by-sa/3.0/deed.en).

(Supplementary Fig. 17). In line with these observations, coordinately upregulated gene modules in human neurons are enriched in promoter CG hypomethylation, whereas downregulated modules are significantly enriched in CH hypermethylated gene bodies (Supplementary Data 15). These results illuminate contrasting yet additive effects of CG and CH during recent evolution of human neurons.

**Developmental and cellular specificity of CH methylation.** CH methylation is nearly absent in fetal brains and accumulates rapidly after birth[1]. We thus hypothesized that the repressive impact of CH methylation might be more pronounced in early postnatal development, and subsequently examined gene expression data from bulk brain tissue during development[14]. Indeed, genes bearing signatures of human-specific CH methylation accumulation (referred to as human CH DMR genes, Supplementary Data 16, Methods) are similarly expressed in

human and macaque brains during prenatal growth but show reduced expression in humans following birth (Fig. 3d). In contrast, chimpanzee CH DMR genes do not exhibit such a pattern (Fig. 3d and Supplementary Fig. 18). We integrated our data with those from sorted neurons from individuals of different ages[2,8], to examine cell-type differences. Human CH DMR-genes showed lower expression in neurons than in non-neurons or oligodendrocytes in most developmental stages, and the reduction of neuronal expression was more evident in toddler and early teen data compared to data from adults (Supplementary Fig. 19).

Interestingly, human CH DMR genes are significantly enriched in gene sets representing inhibitory neurons, based on single-nucleus transcriptome data from the middle temporal gyrus[36] (Fig. 3e), as well as those previously identified as markers of inhibitory neurons[12,37,38] (fold-enrichment = 5.3, $P < 0.0001$, permutation test). Moreover, these genes were more highly methylated in excitatory neurons than in inhibitory neurons in single-nucleus DNA methylation data from the same brain

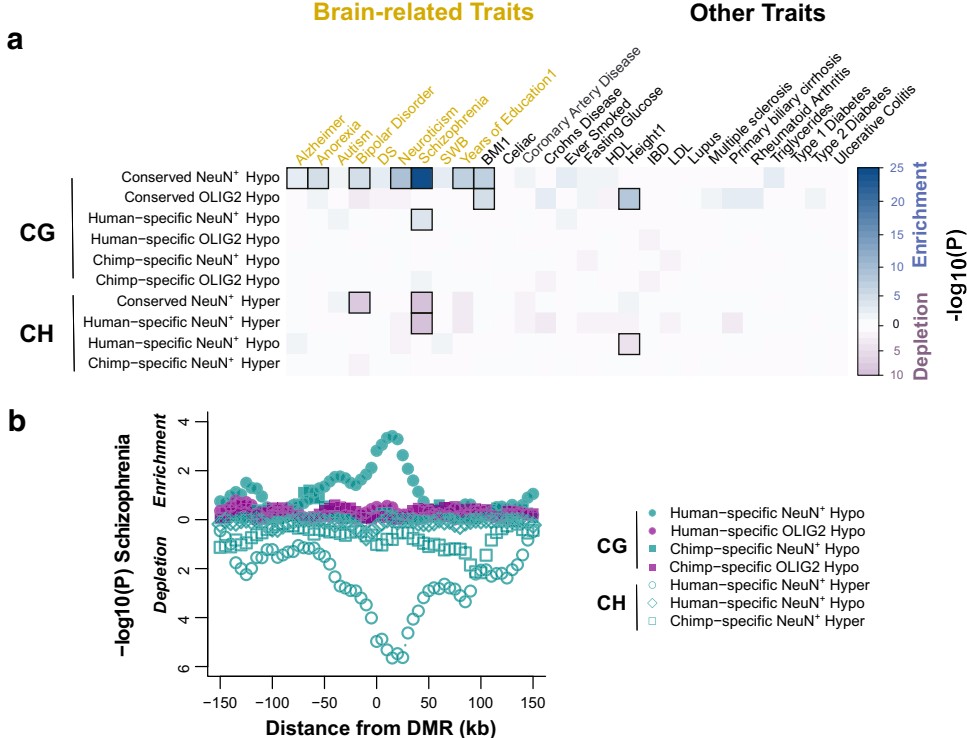

**Fig. 4 Evolutionary DMRs contribute to brain disease susceptibility. a** Significance levels for the enrichment for genetic heritability in different DMRs (+/-25kb) and complex traits. Both conserved and human-specific neuronal DMRs are associated with schizophrenia. Enrichment with FDR < 0.05 are highlighted in squares. Notably, CG DMRs hypomethylated in NeuN+ cells compared to OLIG2+ cells in all three species (conserved NeuN+ hypo) are highly enriched in variants for several brain-related traits, and human-specific NeuN+ hypo shows enrichment in schizophrenia. **b** A sliding-window analysis further demonstrates that the aforementioned signal for schizophrenia was centered at the DMRs and did not originate from extended adjacent regions. The y-axis represents the P-values in sliding windows around DMRs classified by species (human or chimpanzee), cell-type (NeuN+ or OLIG2+), and cytosine context (mCG or mCH). Source data are provided as a Source Data file.

region[12] (Fig. 3f and Supplementary Fig. 20). Integrating these observations, we hypothesize that human-specific CH methylation of inhibitory neuron-specific genes may silence their expression in the genomes of excitatory neurons, thereby promoting functional specificity of neuron subtypes. Alternatively, there may have been a substantial shift of cell-type composition in the human brain since the divergence from chimpanzees, increasing the ratio of excitatory to inhibitory neurons.

**Human neuron-specific CG methylation contributes additional risk to schizophrenia heritability.** We have previously shown that genomic regions exhibiting differential CG methylation between neurons and oligodendrocytes are associated with increased risk for neuropsychiatric disorders, especially for schizophrenia[7]. Other studies have noted that sites of differential histone modification[6] or DNA methylation[33] between neurons and non-neurons (NeuN-cell populations) significantly contribute to heritability for neuropsychiatric disorders. Our data can provide further insights into the evolution of genetic risk for neuropsychiatric disorders.

We used the stratified linkage disequilibrium score regression framework[39] to estimate the contribution of DMRs to the genetic heritability of various diseases and complex traits (Methods). We found a strong enrichment of risk for schizophrenia and other brain-related traits at neuron-hypo CG DMRs that are evolutionarily conserved in the three species, while no signal was detected at OLIG2+ conserved DMRs (Fig. 4a and Supplementary Data 17, 18, and 19). Non-brain polygenic traits such as height and body mass index (BMI) were also detected, consistent with the

previously proposed role of the central nervous system in the genetic architecture of BMI[39]. Moreover, human-specific neuron-hypo CG DMRs exhibited significant enrichment for schizophrenia heritability (Fig. 4a, b), even though the degree of enrichment is lower than that for the conserved DMRs as suggested by down-sampling analyses (Supplementary Fig. 21). In contrast, chimpanzee neuron-hypo CG DMRs did not show significant enrichment for any human trait, while both conserved and human-specific CH DMRs show significant depletion for schizophrenia heritability (Fig. 4a, b). Notably, the depletion signal was centered around the CH DMRs, whereas no other diseases (with the exception of bipolar disorder) nor chimpanzee-specific regions showed a significant trend (Fig. 4a), implying that CH hypermethylated genomic regions are devoid of common DNA polymorphisms associated specifically with schizophrenia. Given that CH DMRs are enriched in inhibitory neuron markers, this observation may suggest that different neuron subclasses contribute disproportionately to schizophrenia phenotype[40,41].

**Discussion**

Decades of research have solidified DNA methylation as a critical regulatory mechanism in human brains, including but not limited to brain development[1,2,42], cell-type differentiation[1,7,12], and disease susceptibility[7,33]. These processes are associated with cognitive and neurodevelopmental programs and neuropsychiatric disorders that are key to human uniqueness[43,44]. Despite such importance for genome regulation and human evolution, how DNA methylation and other epigenetic mechanisms have changed in human brains have not previously been characterized at the cell-type level. Reliable identification of human-specific epigenetic

modifications at the cell-type level has been a limiting factor in previous studies due to the heterogeneity of brain tissue and the different relative cell compositions of different species. Here, we have presented comprehensive analyses of whole-genome methylomes of neurons and oligodendrocytes from humans, chimpanzees, and rhesus macaques, thus elucidating evolutionary changes of DNA methylation during human brain evolution with unprecedented cell-type resolution.

We have previously demonstrated an excess of CG hypomethylation in human prefrontal cortex compared to chimpanzee[4], mostly impacting noncoding regulatory regions of the human genome[5]. We find this to be the case for both neurons and oligodendrocytes, which could contribute to increased gene expression levels that have been reported in human brains[8,45–48]. Furthermore, these epigenomic innovations connect to potential underpinnings in genome evolution. For example, human-derived hypomethylated CG DMRs are enriched for binding motifs for specific transcription factors including FOXP1, a hub gene in human-specific transcriptional networks in the brain and which is implicated in several cognitive diseases in humans, including language, intellectual disability, and autism[49]. In addition, non-coding human accelerated regions (ncHAR) are preferentially found in human-specific hypomethylated CG DMRs. These results begin to reveal the connections between genetic and epigenetic innovations of the human brain involving CG hypomethylation.

Intriguingly, we show that CH methylation is significantly higher in human and chimpanzee neurons in prefrontal cortex compared to those of rhesus macaque and mice. Moreover, neurons in human prefrontal cortex have higher levels of CH methylation than those of chimpanzees and rhesus macaques. Although more data from brains of a wider variety of primates and other mammals are necessary to fully understand evolutionary dynamics of DNA methylation, our observation suggests that CH methylation in the prefrontal cortex neurons has increased during the evolution of primates. CH methylation is highly negatively correlated with gene expression and is a strong predictor of gene-expression divergence between neurons of different species (Fig. 3c and Supplementary Fig. 16). Consequently, the evolutionary trajectory of increasing CH methylation during primate brain evolution may have contributed to shaping finer resolution transcriptional identities of cell types. In this regard, yet a further human-brain specific increase of CH methylation is intriguing. Based on joint analyses of CG and CH methylation, we show that DNA methylation of these distinctive cytosine contexts both contribute additively to the human brain transcriptional program. Integrating our results with developmental bulk tissue data and single-cell functional genomics data from human brains, we show that the human-specific increase of CH methylation appears particularly important for early human brain development, and fine-tuning of neuron subtype cell identities.

Owing to the limitation of bisulfite sequencing, our data cannot separate methylcytosines from hydroxymethylcytosines (hmCs), which might play distinctive roles in neuron subtypes[50]. While additional data are needed, currently available maps[51] do not suggest a significant impact of hmC on the differential methylation patterns identified in this study (Supplementary Fig. 22).

Our data also demonstrate that the majority of differential DNA methylation between neurons and oligodendrocytes has long been established before the divergence of apes and other catarrhine monkeys, echoing that a large portion of human brain regulatory programs have deep evolutionary roots[52]. We further investigated the implication of this finding in the context of a complex neuropsychiatric disorder. We and others have previously shown that, in humans, epigenetic differences between neurons and non-neuronal cells are prevalent in non-coding

regions and locate in regions that account for schizophrenia heritability[6,7,33]. Here, we show that genomic regions with differential CG methylation between neurons and oligodendrocytes that contribute greatest to schizophrenia risk originated before the emergence of the catarrhine ancestor. It is known that genomic regions under strong and ancestral purifying selection (thus remain conserved) are enriched for disease genes and heritability[39,53,54]. For example, ancient enhancers and promoters have greater contributions to susceptibility to complex diseases compared to more recently evolved regulatory regions[54]. Our results suggest that even though the phenotype of schizophrenia is highly specific to humans, the molecular and developmental mechanisms of this disease have deep phylogenetic roots. Moreover, human brain-specific CG hypomethylation provides additional significant genetic risk to schizophrenia, albeit a relatively small proportion. Therefore, recent, human brain-specific epigenetic changes also contribute to schizophrenia pathology. These results advance our understanding of the relevance of conserved and derived regulatory mechanisms to the genetic and epigenetic architecture of complex diseases.

## Methods

**Sample acquisition, whole-genome sequencing, and whole-genome bisulfite sequencing.** Information on samples used in this work was previously described in Berto et al.[8]. Briefly, adult human post-mortem brain samples from Brodmann area 46 (BA46) were acquired from the National Institutes of Health NeuroBioBank (the Harvard Brain Tissue Resource Center, the Human Brain and Spinal Fluid Resource Center, VA West Los Angeles Healthcare Center, and the University of Miami Brain Endowment Bank) and the University of Texas Neuropsychiatry Research Program (Dallas Brain Collection). These samples included 25 and 22 NeuN+ and OLIG2+ specimens, respectively. Non-human primate tissue samples were obtained from Yerkes National Primate Research Center (macaque samples) and the National Chimpanzee Brain Resource (chimpanzee samples). For human samples, UT Southwestern Medical Center Institutional Review Board (IRB) has determined that as this research was conducted using post-mortem specimens, the project does not meet the definition of human subjects research and does not require IRB approval and oversight. Non-human primate samples were obtained from archival, post-mortem brain tissue opportunistically collected from subjects that died from natural causes, and following procedures approved by the Emory Institutional Animal Care and Use Committee and in accordance with federal and institutional guidelines for the humane care and use of experimental animals. No living great apes were used in this study. All non-human primate samples were obtained from homologous regions in chimpanzees (NeuN+ $n = 11$, OLIG2+ $n = 11$) and rhesus macaques (NeuN+ $n = 15$, OLIG2+ $n = 13$).

Nuclei isolation was performed as described previously[7]. Briefly, frozen post-mortem brain was homogenized and subject to sucrose gradient and ultracentrifuge. The resulting nuclei pellet was then incubated with mouse NeuN and OLIG2 antibodies (alexa488 conjugated anti-NeuN (1:200), #MAB377X, Millipore, Billerica, MA and rabbit alexa555 conjugated anti-OLIG2 (1:75), #AB9610-AF555, Millipore). We then performed the fluorescence-activated nuclei sorting (FANS), followed by nucleic acid purification via the ZR-Duet DNA/RNA MiniPrep (Plus) kit (#D7003, Zymo Research, Irvine, CA) (Supplementary Fig. 23).

**Whole-genome bisulfite data processing.** We followed the same data processing steps described in our previous work[7]. Briefly, extracted DNA was fragmented by S-series Focused-ultrasonicator (Covaris, Woburn, MA) using the "200 bp-target peak size protocol". Fragmented DNA was then size selected (200–600 bp) with an Agencourt AMPure XP bead-based (#A63880, Beckman Coulter, Brea, CA), followed by the End repair step was performed with End-It DNA End-Repair Kit (#ER81050, Epicenter, Madison, WI) and A-tailing (#M0202, New England Biolabs, Ipswich, MA), and ligation of methylated adaptors (#511911, B100 Scientific, Austin, TX). The methylome libraries were diluted and loaded onto Illumina HiSeqX system for sequencing using 150 bp paired-end reads. We performed quality and adapter trimming using TrimGalore v.0.4.1 (Babraham Institute) with default parameters. Reads were mapped first to PhiX genome (NC_001422.1) to remove the spike-in control and the remaining reads were subsequently mapped to the chimpanzee PanTro5 and macaque rheMac8 reference genomes using Bismark v 0.14.5[55] and bowtie v2.3.4[56]. After de-duplication, we obtained coverage for over 84% of the CpGs in the chimpanzee genome with an average read depth 19.32x, and over 91% of CpGs in the macaque genome with an average read depth of 21.61x. We calculated fractional methylation (ratio of the number of methylated cytosine reads to the total number of reads) levels at individual cytosines. Bisulfite conversion rates were estimated by mapping the reads to the lambda phage genome (NC_001416.1).

**Whole-genome sequencing data processing.** Quality and adapter trimming was performed using TrimGalore v.0.4.1 (Babraham Institute) with default parameters. Reads were mapped to the hg19, PanTro5 or rheMac8 reference genomes using BWA v0.7.4[57] and duplicates were removed using picard v2.8.3 (https://broadinstitute.github.io/picard/index.html). We identified genetic polymorphisms from re-sequencing data following the GATK v4 best practices workflow[58]. For base recalibration, we used vcf files for known variants from dbSNP for chimpanzee and macaque from the following links: ftp://ftp.ncbi.nlm.nih.gov/snp/organisms/chimpanzee_9598/VCF/ and ftp://ftp.ncbi.nlm.nih.gov/snp/organisms/macaque_9544/VCF/. We applied hard filters for genotype calling with the following parameters: --filterExpression "QD < 2.0 || FS > 60.0 || MQ < 40.0 || MQRankSum < −12.5 || ReadPosRankSum < −8.0". For chimpanzee, we identified 10,980,856 variants with mean depth >24x. For macaque, we identified 30,001,119 variants with mean depth >24x. Since C > T and G > A polymorphisms at CpG sites can generate spurious differential methylation patterns, we removed polymorphic CpGs from downstream differential methylation analyses keeping a total of 26,024,877 and 24,740,404 non-polymorphic CpGs for chimpanzee and macaque genomes, respectively. For quality control of SNP calling, we performed principal component analyses using additional chimpanzee and bonobo samples from de Manuel et al.[59] using 75,575 common SNPs from chromosome 20. As expected, our chimpanzee samples clustered with other chimpanzees and not with bonobos (Supplementary Fig. 24). We recapitulated the genetic ancestry of de Manuel et al. samples and identified most of our individuals as Western chimpanzees (*Pan troglodytes verus*) while one sample (sample ID Anja) clustered with Nigeria-Cameroon chimpanzees (*Pan troglodytes ellioti*).

**Transcription factor motif enrichment analyses.** We performed TF enrichment tests using the MEME suite's[60] AME software and two HOCOMOCO v11 databases[61] of human TF motifs. We used seven primates (human, chimpanzee, gorilla, orangutan, rhesus macaque, baboon, and gibbon) for which we have high-quality genome sequences to identify cytosines that are conserved in all seven primate species ($n = 567,893$) as "conserved CpGs". In comparison, "variable CpGs" refer to CpGs that are specific to humans but not in other primates ($n = 237,956$). We identified TF motifs enriched at variable CpGs compared to conserved CpGs, as defined above. For this analysis, we added 20 bps to each side of each CpG given that the longest motif length in the database is 25 bp. We compared the variable CpGs to control CpG sets as follows. We ran AME 100 times comparing the variable CpGs to a matched number of random CpG (defined as not overlapping with variable and conserved CpGs) using the following command:

ame --verbose 2 --oc variable_CpG.fa --scoring avg --method fisher --hit-lo-fraction 0.25 --evalue-report-threshold 10.0 --control control_CpG_1.fa HOCOMOCOv11_core_HUMAN_mono_meme_format.meme

Similarly, we also ran AME for conserved CpGs using 100 control CpG sets as background, as well as using the Full Homococo v11 database.

We subsequently defined variable CpG-specific motif as those that satisfy both of the following conditions:

(frequency of enrichment in variable CpGs compared to control CpGs > 0.95 in the 100 comparisons) AND (frequency of enrichment in conserved CpGs compared to control CpGs < 0.05).

In comparison, conserved CpG-specific motifs are those that satisfy both of the following conditions:

(frequency of enrichment in variable CpGs compared to control CpGs for >0.95) AND (frequency of enrichment in conserved CpGs compared to control CpGs < 0.05).

A total of 81 and 121 motifs were identified as variable CpG-specific and conserved CpG-specific in the core database, and 183 and 190 in full database, respectively (Supplementary Data 5). The TF families with at least 5% difference between the two categories are shown in Supplementary Fig. 6.

We also applied MEME suite's AME software and two HOCOMOCO v11 databases to compare human-hypomethylated DMRs to chimpanzee-specific hypomethylated DMRs. We extended the DMRs 10 bp to each side and run AME using the parameters as shown before. We found three TF motifs significantly associated with human hypomethylated DMRs, including two Forkhead box factors (FOXP1 and FOXK1) and the nuclear factor 1 C-type, NFIC. Identical results were obtained for both core and full datasets. 79% of human-hypomethylated DMRs showed a hit in any of the three TF motifs. A total of 1996 human-specifically hypomethylated DMRs associated with FOXP1 motif, 1906 DMRs with FOXK1 and 462 with NFIC motif. The DMRs with positive hits were highly shared among TFs, with around 80% shared between FOXP1 and FOXK1, and around 60% of NFIC binding-DMRs also bind the other two TFs. We compared the methylation levels of these DMRs and the associated gene expression patterns compared to other DMRs without enriched motifs (Supplementary Fig. 13).

**RNA-Seq data.** We used our previously generated matched samples of RNA-Seq datasets for human (without brain-related diseases), chimpanzee, and rhesus macaques from GSE108066, GSE107638, and GSE123936. The list of differentially expressed genes (DEGs) were also obtained from this previous work[8].

**Liftover of non-human primates cytosine positions to human genome.** We lifted over the non-human primates' cytosine coordinates to human hg19 genome using UCSC batch liftover tool (panTro5ToHg19.over.chain.gz and rheMac8ToHg19.over.chain.gz for chimpanzee and rhesus macaque, respectively). For the CG DMR analysis, we did not perform three-way species analyses based on lifted over coordinates due to the rapid evolutionary loss of CG sites since the macaque split. Compared to around 21 million CG sites conserved between human and chimpanzee, only around 9.6 million CGs are conserved between human and macaque, whereas 13 million CGs in macaque show non-CG dinucleotides in human. To circumvent this issue, we first identified human-chimpanzee differentially methylated regions (DMRs) using conserved CGs and then used orthologous regions in the macaque rheMac8 genome to polarize the DMRs (see "*Incorporation of Rhesus Macaque as an outgroup species*" for additional details). We removed cytosines located in paralogous sequences in at least one species to avoid erroneous mapping (i.e., one-to-many or many-to-one mapping between species). For the CH methylation analysis, we used orthologous cytosines conserved among the three species.

**Identification of CG differential methylation.** We identified differentially methylated positions of (1) cell-types (NeuN+ vs. OLIG2+), (2) species (human vs. chimpanzee where both cell types show the same direction and magnitude of methylation differences between two species), and (3) cell-type-specific species changes (either cell-type exclusively shows DNA methylation difference between species) using DSS (ver. 2.3) Bioconductor package[62]. DSS handles variance across biological replicates and models read counts from WGBS experiments while accounting for additional biological factors. Specifically, we considered age (converted to three level categorical variable), sex, and conversion rates as covariates in the following model;

Fractional methylation ~ cell_type + species + species:cell_type + sex + age_class + conversion_rates

To remove low coverage loci, we only included sites with at least 5x coverage in 80% of individuals per species or cell-type. We used a false discovery rate (FDR) threshold of 5% to identify significant differentially methylated positions. For DMR identification, we considered a minimum length of 50 bp with at least four significant differentially methylated positions. We removed cell-type DMRs and species DMRs that overlap with cell-type-specific species changes (i.e., interaction of cell-type and species effects) to remove redundant DMRs. We only considered the DMRs that show >10% of average methylation difference between human and chimpanzee for species DMR and >15% of average methylation difference between cell-types for cell-type DMR (please also see the section "*Incorporation of Rhesus Macaque as an outgroup species*" for detailed explanation of final set of DMRs).

Of note, as our differential methylation analyses were run under a multifactor design in DSS, the estimated coefficients in the regression were based on a generalized linear model framework using the arcsine link function to reduce dependence of variance on the fractional methylation levels[63]. The distribution of the statistic is determined by the differences in methylation levels, as well as by biological and technical factors such as read depth. The sign of the test statistic indicates the direction of methylation. However, the values of the test statistic cannot be directly interpreted as fractional methylation differences. For DMRs, the tool generates "areaStat" values, which are defined as the sum of the test statistic of all CG sites within the DMR. To identify the stringent sets of DMRs we excluded DMRs if the average test statistics of corresponding CGs in the region (areaStat divided by the number of CGs) was below the test statistic corresponding to FDR = 0.05.

**Incorporation of rhesus macaque as an outgroup species.** We retrieved the corresponding genomic coordinates in rheMac8 using the Ensembl Primate EPO multiple sequence alignment[64]. Read counts and methylation values of the CGs in corresponding regions were obtained from the macaque samples. Only CG sites with at least 5x coverage in 80% of the individuals per species were considered. The DMRs resulting from human and chimpanzee samples that had low alignment coverages with macaque (<50%) or included <4 CGs in macaque were considered "unclassified" DMRs. After adding macaque data, we fitted a beta regression model using the average methylation level of each individual accounting for the covariates indicated above. Among the cell-type DMRs resulting from human and chimpanzee samples, DMRs in which macaque showed cell-type changes in the same direction and exhibited >15% fractional methylation difference were considered conserved cell-type DMRs.

We then used stringent criteria to categorize the species specificity of DMRs as human- or chimpanzee-specific. For example, a human-specific hypomethylated DMR should satisfy the following criteria: (1) the average fractional methylation of human is significantly lower than that of chimpanzee and macaque (FDR < 0.05), (2) the absolute methylation difference between human and macaque is greater than that between chimpanzee and macaque, (3) the proportion of the absolute methylation difference between human and macaque is >5%, and (4) both of the two cell-types satisfy these criteria. Those DMRs that did not satisfy these criteria were considered "unclassified". We used the same logic to specify human-specific hypermethylated DMRs and chimpanzee-specific hypo- and hypermethylated DMRs. We also examined species-specific DMRs that show differential methylation between species but exclusively in one cell-type (i.e., either cell-type

shows differential methylation patterns derived from either the human or chimpanzee lineage).

**Identification of CH differential methylation**. Unlike CG methylation, >70% of cytosine positions were conserved among the three species. Thus, we used orthologous cytosines across the three species to infer differentially methylated positions. As CH methylation is sensitive to bisulfite conversion rate[65], we only used individuals with high bisulfite conversion rates (>99.5%). We down-sampled and matched sample size across the species to avoid any bias derived from the different sample sizes across groups ($N = 11$ for each species and cell-type). We removed sites in which >50% of individuals in at least one group have fewer than five read counts.

For each CH site, we fitted a generalized linear model using the arcsine function to identify differentially methylated CH positions among species adjusting for other covariates (age, sex, and bisulfite conversion rate) using DSS. To fit our parsimonious approach, we also performed pair-wise analyses between species considering all combinations (i.e., human vs. chimpanzee, human vs. macaque, and chimpanzee vs. macaque). Benjamini–Hochberg correction (FDR) was used to perform multiple comparisons. We used the parsimonious approach to detect species-specific methylation changes with a cutoff of fractional methylation difference between species >10% and FDR < 0.05. For example, human-specific CH methylated sites showed FDR < 0.05 from both human vs. chimpanzee and human vs. macaque comparisons and FDR > 0.05 from the chimpanzee vs. macaque comparison, as well as a > 10% difference of fractional methylation in humans compared to both chimpanzee and macaque fractional methylation levels.

To identify human-specific and chimpanzee-specific CH DMRs, we identified significantly differentially methylated regions between human and chimpanzee using the differentially methylated positions generated from the human-chimpanzee comparison. To be considered as a CH DMR, the region should be a minimum length of 50 bp harboring at least four significant differentially methylated positions (FDR < 0.05) and covering >10 cytosines. We also applied an average methylation difference of 10% as a cutoff. Using average methylation of macaque from corresponding regions, we detected human-specific and chimpanzee-specific CH DMRs using the following criteria. Human-specific CH DMRs are defined as DMRs that show a significant human-chimp difference with at least four differentially methylated positions, as well as a methylation difference between human and macaque of >5% that is also greater than the methylation difference between chimpanzee and macaque. Similarly, chimp-specific CH DMRs are DMRs that satisfy the following criteria: a significant human-chimp difference with at least four differentially methylated positions and a methylation difference between chimpanzee and macaque of >5% that is also greater than methylation difference between human and macaque. To obtain regions in which both human and chimpanzee were differentially methylated compared to macaque, we checked the overlap between human-macaque CH DMRs and chimpanzee-macaque CH DMRs.

**Identification of DMR genes**. To identify differentially methylated genes, we extracted genes with at least one DMR within a 3 kb window upstream and downstream of the gene body. To remove redundant genes among different categories of DMR genes, we used average gene body methylation as an additional indicator to assign genes into the DMR gene category using the following criteria. Human-specific hyper CH DMR-genes are defined as DMR genes that include at least one human-specific hyper CH DMR and show higher average gene body methylation compared to the average gene body methylation of chimpanzee and macaque. Also, the absolute methylation difference between human and macaque should be greater than the methylation difference between chimpanzee and macaque.

**CH methylation of neuronal subtypes**. We examined methylation patterns of neuronal subtypes for CH DMR genes. Average gene body methylation of CH DMR genes was calculated for neuronal cells from 21 human neuronal subtypes[12]. For the marker gene analysis of neuron subtypes, we used known excitatory and inhibitory neuron markers from Luo et al.[12]. We included the marker genes that are orthologous to the three species. These include 20 excitatory neuron markers (*SATB2, TYRO3, ARPP21, SLC17A7, TBR1, CAMK2A, ITPKA, ABI2, RASAL1, FOXP1, SLC8A2, SV2B, PTPRD, LTK, LINGO1, NRGN, NPAS4, KCNH3, BAIAP2, ARPP19*) and 13 inhibitory neuron markers (*ERBB4, GAD1, SLC6A1, CCNE1, EPHB6, KCNAB3, LPP, TBC1D9, DUSP10, KCNMB2, UBASH3B, MAF, ANK1*).

**Lineage-specific accelerated non-coding regions**. We used a set of human accelerated regions from Capra et al.[29], which combined regions identified from independent studies (i.e., the 721 "Pollard HARs" from Lindblad-Toh et al.[66], the 1356 "ANC" regions from Bird et al.[67], the 992 "HACNS" regions from Prabhakar et al.[68], and the 63 "Bush08" regions from Bush and Lahn[69]). Statistical significance and fold-enrichment for DMRs were computed from the occurrences of DMRs for each feature compared to GC-matched control region sets ($n = 100$).

**Hydroxymethylation**. We used previously published methylome and hydroxymethylome maps at nucleotide resolution in the adult human brain[51]. The hmC

and mC sites were defined in the original paper. We included the cytosines that are orthologous across the three species ($n = 2,905,389$). We compared the proportions of differentially methylated loci between 5-hydroxymethylcytosines (hmC) and 5-methylcytosines (mC). The proportions of the differentially methylated loci at hmC loci (4.2%) and mC loci (4.2%) showed no difference.

**Contribution of DMRs to disease heritability using stratified LD score regression**. To quantify the contribution of DMRs to the genetic risk of different traits and diseases, we performed stratified LD score regression analyses[39]. This method estimates the percentage of heritability explained by a set of SNPs in a certain trait using GWAS summary statistics and computes the enrichment and significance by comparing the observed heritability to the expectation given the fraction of the genome considered. We used default parameters and excluded the MHC region as in Finucane et al.[39]. Together with the DMR annotations, we also included the basal functional categories described in the original paper. The list of GWAS traits and references are listed in Supplementary Data 11.

The stratified LD score regression method produces large standard errors when the annotation categories cover a small fraction of the genome. Since evolutionary DMRs are generally short (e.g., the median lengths of human CG DMR and CH DMR are 471 bps and 246 bps, respectively) we extended the DMR windows by 25 kbp on both sides to improve the confidence intervals of the estimates as in other studies[70]. To ensure the GWAS signals were centered around the DMRs and not emerging from the extended regions, we further performed the stratified LD score regression in sliding windows 300 kb around the DMRs with a window size of 20 kb and step size of 5 kb.

Conserved CG DMRs were more numerous and longer than human-specific ones, which could lead to increased statistical power on stratified LD score regression analyses. In order to directly compare the significance of conserved and human-specific DMR categories to schizophrenia heritability, we performed partitioned stratified LD score analyses using 100 random sub-samplings of conserved regions. As shown in Supplementary Fig. 16, even when comparable datasets were used, conserved neuron hypomethylated DMRs consistently showed larger enrichments than human-derived ones. Similarly, subsampling of human hyper CH DMR to chimpanzee hyper DMR number and length showed stronger depletion in the former (Supplementary Fig. 16). These analyses indicate that the observed differential patterns are not due to different DMR number and lengths.

We provide a list of human-specific neuron hypomethylated DMRs and conserved neuron hypomethylated DMRs harboring credible schizophrenia-associated SNPs ($P < 10^{-5}$) in Supplementary Data 19. We report evolutionary DMRs at some of the top GWAS signals, including the MHC region (excluded from LDSC analyses) and *CACNA1C* gene among others.

**Reporting summary**. Further information on research design is available in the Nature Research Reporting Summary linked to this article.

## Data availability

Human TF motifs data were downloaded from HOCOMOCO v11 database (https://hocomoco11.autosome.ru/). Single-cell methylome data were downloaded from GEO (https://www.ncbi.nlm.nih.gov/geo/query/acc.cgi?acc=GSE97179). Single-cell transcriptome data were downloaded from Allan Brain Atlas database (https://portal.brain-map.org/atlases-and-data/rnaseq). All raw data and processed methylation data described in this study have been deposited in the NCBI Gene Expression Omnibus and available at GEO Series accession number GSE151768. Source data are provided with this paper.

## Code availability

The code for processing methylation data is available at Github at https://github.com/soojinyilab/Brain_methylome_NHP.

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

## Acknowledgements

This work was partially supported by the Asan Foundation (Biomedical Science Scholarship) to H.J.; Uehara Memorial Foundation to N.U.; JSPS Grant-in-Aid for Early-Career Scientists (18K14814) to N.U. and Scientific Research (C) (18K06977) to K.T.; Takeda Science Foundation to N.U.; the JSPS Program for Advancing Strategic International Networks to Accelerate the Circulation of Talented Researchers (S2603) to S.B., N.U., K.T., and G.K.; the James S. McDonnell Foundation 21st Century Science Initiative in Understanding Human Cognition —Scholar Award and the Jon Heighten Scholar in Autism Research at UT Southwestern to G.K.; National Science Foundation (SBE-131719 and EF-2021635) to S.V.Y; and the NIMH (MH103517), to T.M.P., G.K., and S.V.Y. The National Chimpanzee Brain Resource was supported by NINDS (R24NS092988). Macaque tissue collection and archiving was supported by the NIH National Center for Research Resources (P51RR165; superseded by the Office of Research Infrastructure Programs (OD P51OD11132)) to the Yerkes National Primate Research Center.

## Author contributions

Conceptualization and funding—T.M.P., G.K., and S.V.Y.; sample acquisition and dissection—G.K., T.M.P., and S.V.Y.; nuclei sorting—N.U., K.T., and C.D.; WGBS, WGS data generation: P.C., and T.L.; data curation and bioinformatics assistance—D.S., S.B., I. H.; bioinformatics analysis—H.J., I.M., G.K., and S.V.Y.; drafting—H.J., I.M., and S.V.Y.; writing—H.J., I.M., G.K., T.M.P., and S.V.Y.

## Competing interests

The authors declare no competing interests.
