## [Peer Review File · Nature Communications]

Reviewers' Comments:

Reviewer #1:

Remarks to the Author:

This is a very comprehensive paper on differential DNA CpG and CpH methylation on a genome-wide scale across three different primate species, being the first to examine this in cell-type specific manner with two major cell types NeuN+ and Olig2+ nuclei in adult prefrontal cortex.

With altogether approximately 100 datasets generated, with double digit number of assays even for chimpanzees (a species which has very limited brain tissue availabilities in the present times), the paper will provide a unique, and at the same time, extremely critically needed, resource for the field. From this point. I am very enthusiastic about this paper.

Methodology, and their genomic studies as put forward in this paper, looks state-of-the-art, to the best of my knowledge.

My only criticism refers to their (at face value) extremely interesting observation that human specific CpH methylation signals are primarily associated with inhibitory genes. The Authors propose, if I understood correctly, that this signal is derived from excitatory neurons in the cortex.

While the Authors hypothesis is probably correct, the fact that they sorted NeuN+ and not excitatory or inhibitory subtypes specifically, does not allow a firm conclusion. Even more importantly, could there be a shift in cell type composition with altered glu/gaba neuron nuclei proportions across the cortical tissue samples, due to species-specific differences or other confounds? In a revised manuscript version, these issues should be discussed more carefully, if it is not able to test directly.

Another important criticism, even if it is only 'minor', is that the abstract, as written in its current form, is overly vague, with repetitive wording at times, and not reflective of the rich findings and datasets presented in the paper. This paper deserves a better abstract.

Reviewer #2:

Remarks to the Author:

DNA methylation shows strong dynamics during mammalian brain development and cell-type specific patterns, the comparative study of DNA methylation across primate brains is highly relevant to further the understanding of human brain evolution. This study has generated a valuable resource of whole-genome bisulfite sequencing (WGBS) for two brain cell populations from chimpanzees and macaques. The WGBS dataset from purified cell populations in non-human primates and with large sample sizes n=11 (for WGBS) is clearly novel. The overall quality of computational analyses is sound. However, there are several important areas of analyses that were not explored in this study and should be improved before publication.

1.The low rate of conservation of CpG sites between macaque is an interesting observation and potentially has broad implications in gene regulation. Are CpG sites significantly more diverged compared to other cytosine contexts (CH sites)? I would suggest the authors present an extended CpG and CH site conservation analysis for other non-human primates (both old world and new world monkeys) used in research such as marmoset, lemurs, etc, and also show mouse as an outgroup.

2.What is the genomic distribution of conserved or variable CpG sites, relative to known functional annotations such as CGIs, enhancers, etc.?

3.Some transcription factors (TFs) contain CpG sites in their binding sites. Their binding may also be regulated by the methylation states as shown in in vitro binding studies such as methylSELEX (Yin et al., Science 2017). Does the divergence of CpG sites among primates preferentially affect the binding site of certain transcription factors? Or conversely could the binding of certain TF family to CpG sites

prevents methylation and protects the sites from nature deamination?

4. Human-specific hypo CG-DMRs is a very important finding from this study. Although some examples were discussed, the current manuscript did not provide a clear answer to the question of whether these human specific DMRs were predominantly driven by human-specific gene overexpression, as shown in the case of CLUL1, or local sequence variants (i.e. methylQTL) also have a significant contribution. There may be several ways to address this question - first the authors should analyze the location of these DMRs. If these DMRs are primarily driven by gene overexpression in human, then likely most human specific DMRs are located within gene bodies and enriched at the immediate downstream of TSS. Alternatively, if many of the human specific DMRs are clearly distal regulatory elements, then these DMRs may indicate certain human-specific regulatory program, which could be identified by TF binding motif analysis.

5. How many variants across the three primate species also overlap with human methyl-QTL sites? I went back to an earlier paper by the same group of authors (Mendizabal et al., 2019, Genome Biology). It was somewhat surprising that no methyl-QTL analysis was performed in the earlier work. The question here is whether the DMRs showing variable methylation across primates are also variable across the human population and whether there is a genetic (sequence) underpinning.

6. When analyzing gene body CH methylation, the authors have preliminary used the raw CH methylation ratio instead of normalizing to global CH methylation level of each species. This would by default leads to more hyper-CH methylated genes in the human. Do the conclusions in Fig. 3 holds if the authors use normalized CH methylation levels (normalized against genome average) for the analysis?

7. The results shown in Fig.2 and Fig.3 are somewhat contradictory to one another, which could be due to that raw instead of normalized CH methylation level was used in Fig.3. Presumably if the authors find human specific CG-DMRs overlapping with highly expressed genes such as CLUL1, the gene should also have a lower CH methylation level due to the inverse correlation between CH methylation and gene expression. In any case, it would be good to clarify whether the gene sets whose expression showing a strong correlation with CG and CH methylation are overlapped or not.

8. The partitioned heritability analysis is standard for this type of study and is also informative. Can the authors provide an example and a list of the overlaps between schizophrenia credible SNPs (from finemapping) and DMRs?

Reviewer #3:

Remarks to the Author:

In "Evolution of DNA Methylation in the Human Brain", the authors generate DNA methylation data from neurons and oligodendrocytes in the prefrontal cortex of human, chimp, and macaque. They identify conserved and species-specific sites of mCG and mCH and increased mCH and decreased mCG in the human lineage. They increase the interpretability of their finding by integrating their data with several publicly available datasets. They show that these methylation changes may act to change gene expression at specific developmental timepoints and in a specific classes of cortical neurons. Surprisingly, they find that genomic loci associated with many brain disorders and traits are enriched in conserved mCG sites and only schizophrenia loci were enriched in human-specific mCG sites. This paper provides a useful dataset to the scientific community interested in comparative genomics and human brain disease. The results would be strengthened by adding a power analysis to show that differences in the enrichment of brain traits for different sets of mCH and mCG sites are not driven by differences in the number of risk loci and/or sites.

Specific comments:

- SI Fig. 5 – Please move these data to main Fig. 1F and test whether there is a significant difference between human and chimp. If not, this suggests that evolutionary rates are similar along the two lineages.
- Discussion – Please comment on the possibility that some species-specific differences in the NeuN+ population may be driven by evolutionary changes in proportions of different cell types in DLPFC. For example, a potentially greater expansion of supragranular neurons in human vs. chimp.
- Fig. 3d/SI Fig. 13 – Macaque CH DMR genes have lower expression in macaque than human throughout development. Please comment on this difference during fetal development vs. human CH DMR genes. What are the results of a similar analysis using human CG DMR genes? Do you see lower expression of human than macaque during prenatal development, suggesting that this CG methylation persists through development?
- Fig. 4A – Could the lack of significant enrichment for diseases in species-specific mCH/mCG be due to a lack of power? If you downsample conserved NeuN+ hypo mCG sites to be the same number as human-specific sites, do you still find significant enrichment for brain diseases? It is surprising that a disease that affects only humans has risk loci found in conserved sites. How do you interpret this? Likewise, how does the number of risk loci compare across diseases? SCZ GWAS have more subjects than many other traits/diseases, and the disease specificity you see in human-specific sites may be driven by increased power to detect enrichment.
- Fig. 4B – Does the depletion of SCZ SNPs in human-specific mCH hyper sites support a role for inhibitory neurons in the disease given your finding that these sites are associated with inhibitory marker genes?
- Please comment on the extent to which differences in genome annotation quality could affect your quantification of DNA methylation and conclusions about excess/depletion of mCG/mCH hyper/hypomethylation. Previous work has suggested that human neurons express more genes than chimpanzee neurons, but a recently improved chimpanzee genome annotation has resulted in more accurate transcript quantification and suggests that overall transcript levels are quite comparable (see Pollen et al. 2019 Cell).

Trygve Bakken

RESPONSE TO REVIEWERS' COMMENTS

Reviewer #1 (Remarks to the Author):

This is a very comprehensive paper on differential DNA CpG and CpH methylation on a genome-wide scale across three different primate species, being the first to examine this in cell-type specific manner with two major cell types NeuN+ and Olig2+ nuclei in adult prefrontal cortex. With altogether approximately 100 datasets generated, with double digit number of assays even for chimpanzees (a species which has very limited brain tissue availabilities in the present times), the paper will provide a unique, and at the same time, extremely critically needed, resource for the field.

From this point. I am very enthusiastic about this paper.

Methodology, and their genomic studies as put forward in this paper, looks state-of-the-art, to the best of my knowledge.

My only criticism refers to their (at face value) extremely interesting observation that human specific CpH methylation signals are primarily associated with inhibitory genes. The Authors propose, if I understood correctly, that this signal is derived from excitatory neurons in the cortex. While the Authors hypothesis is probably correct, the fact that they sorted NeuN+ and not excitatory or inhibitory subtypes specifically, does not allow a firm conclusion. Even more importantly, could there be a shift in cell type composition with altered glu/gaba neuron nuclei proportions across the cortical tissue samples, due to species-specific differences or other confounds? In a revised manuscript version, these issues should be discussed more carefully, if it is not able to test directly.

Response: We appreciate the encouragement and thoughtful comment by the reviewer. We agree that it is difficult to draw a firm conclusion regarding neuronal subtypes without explicit cell-type data from these classes. We have thus revised the manuscript to clearly indicate that a substantial shift of cell type composition between species is also consistent with the observed pattern and that ultimately, this question needs to be re-visited with single cell resolution data in the future. We are actively pursuing that direction of research.

Another important criticism, even if it is only 'minor', is that the abstract, as written in its current form, is overly vague, with repetitive wording at times, and not reflective of the rich findings and datasets presented in the paper. This paper deserves a better abstract.

Response: We appreciate the thoughtful input by the reviewer. We have rewritten the abstract to better represent our work.

Reviewer #2 (Remarks to the Author):

DNA methylation shows strong dynamics during mammalian brain development and cell-type specific patterns, the comparative study of DNA methylation across primate brains is highly relevant to further the understanding of human brain evolution. This study has generated a valuable resource of whole-genome bisulfite sequencing (WGBS) for two brain cell populations from chimpanzees and macaques. The WGBS dataset from purified cell populations in non-human primates and with large sample sizes $n=11$ (for WGBS) is clearly novel. The overall quality of computational analyses is sound. However, there are several important areas of analyses that were not explored in this study and should be improved before publication.

1. The low rate of conservation of CpG sites between macaque is an interesting observation and potentially has broad implications in gene regulation. Are CpG sites significantly more diverged compared to other cytosine contexts (CH sites)? I would suggest the authors present an extended CpG and CH site conservation analysis for other non-human primates (both old world and new world monkeys) used in research such as marmoset, lemurs, etc, and also show mouse as an outgroup.

Response: We thank the reviewer for this constructive comment. As a complement to the low rate of CpG conservation that we presented in the manuscript, we have now added more genome sequences of 10 primate species (human, chimpanzee, gorilla, orangutan, rhesus macaque, baboon, marmoset, tarsier, mouse lemur, bushbaby) including both old world and new world monkeys. Specifically, to examine to which extent the sequence conservation between CG and CH sites differs, we used PhastCons and PhyloP, which are two of the most popular tests to measure evolutionary conservation scores. We used these two different scores as a cross-check because PhyloP measures conservation at individual positions whereas PhastCons scores are affected by neighboring sites (calculated by a window-based approach).

Specifically, we classified all cytosines into four different cytosine contexts based on the subsequent base (i.e. CA, CC, CG, and CT). For each cytosine context, we computed a mean conservation score of 100,000 randomly selected cytosine sites. Then, we repeated this process 1,000 times and compared the mean conservation scores across cytosine contexts. As shown in Response Figure 1, the cytosine context of CG was significantly more diverged ($P < 0.001$) compared to the other cytosine contexts of both PhastCons and PhyloP scores. We also did the same analysis using 33 placental mammals (including mouse, Response Figure 1) to confirm that the results are consistent. These results are now added as Supplementary Figure 2 in the revised article.

Response Figure 1 (also shown as **Supplementary Figure 2** in the revised article). Comparison of evolutionary conservation scores demonstrates rapid divergence of CG contexts. The Y-axis represents the mean conservation score resulting from the cytosine of 100,000 randomly selected positions in each context. Each boxplot displays the mean conservation scores of 1,000 trials.

2. What is the genomic distribution of conserved or variable CpG sites, relative to known functional annotations such as CGIs, enhancers, etc.?

Response: Following the reviewer's comment, we investigated the genomic distribution of the conserved and variable CpG sites. Due to the rapid loss of CpG sites across species, we excluded new world monkeys and used 7 primates (human, chimpanzee, gorilla, orangutan, rhesus macaque, baboon, and gibbon) for which we have high quality genome sequences. To correlate with the functional annotation from the human genome, we used cytosines that are conserved in all 7 primates species (n = 567,893) as 'conserved CpGs'. In comparison, 'variable CpGs' refer to CpGs that are specific to humans and not found in other primates (n = 237,956). Statistical significance and fold-enrichment were computed from the occurrences of the CpGs for each genomic feature in random control sets (n=100). Genomic coordinates of the CpG islands were downloaded from UCSC genome browser. We used genomic coordinates of brain enhancers from the dorsolateral prefrontal cortex using the ChromHMM model downloaded from Roadmap epigenomics.

Conserved CpGs were most significantly enriched in CpG islands (fold-enrichment = 23.24). Conserved CpGs were also highly enriched in exons (fold-enrichment = 10.2) and enhancers (fold-enrichment = 2.26). These results are consistent with the hypomethylation of CpG islands preventing the transition mutations at those sites^{1,2}, as well as sequence constraint on protein-coding sequences on CpG conservation³. On the other hand, variable CpG sites did not exhibit strong enrichment patterns. We included these results in the revised manuscript as Supplementary Figure 5.

Response Figure 2 (also shown as **Supplementary Figure 5** in the revised article). Distribution of evolutionarily conserved CpGs and variable CpGs in different functional genomic regions. Fold-enrichment was computed from the occurrences of the CpGs for each feature compared to random control sets (n=100). Red dashed lines indicate fold-enrichment values of 1.

3. Some transcription factors (TFs) contain CpG sites in their binding sites. Their binding may also be regulated by the methylation states as shown in *in vitro* binding studies such as methylSELEX (Yin et al., Science 2017). Does the divergence of CpG sites among primates preferentially affect the binding site of certain transcription factors? Or conversely could the binding of certain TF family to CpG sites prevent methylation and protect the sites from natural deamination?

Response: Following the reviewer's suggestion, we performed TF enrichment tests using the MEME suite⁴'s AME software and two HOCOMOCO v11 databases⁵ of human TF motifs. We identified TF motifs enriched at variable CpGs compared to conserved CpGs, as defined above. For this analysis, we added 20 bps to each side of each CpG given that the longest motif length in the database is 25bp. Specifically, we ran AME 100 times comparing the variable CpGs to a matched number of control CpGs (defined as not overlapping with variable or conserved CpGs). Similarly, we also ran AME for conserved CpGs using 100 control CpG sets as background.

From these results, we defined variable CpG-specific motifs as those that satisfy both of the following conditions:
 (frequency of enrichment in variable CpGs compared to control CpGs > 0.95 in the 100 comparisons) AND (frequency of enrichment in conserved CpGs compared to control CpGs < 0.05)

In comparison, conserved CpG-specific motifs are those that satisfy both of the following conditions:

(frequency of enrichment in variable CpGs compared to control CpGs for >0.95) AND (frequency of enrichment in conserved CpGs compared to control CpGs < 0.05).

A total of 81 and 121 motifs were identified as variable CpG-specific and conserved CpG-specific in the core database, and 183 and 190 in full database, respectively (Supplementary Table 5 in the revised manuscript). TF motifs observed almost exclusively at variable CpGs belong to FOX, HOX, SOX, POU and interferon-regulatory factors, whereas these families are rare among conserved CpG-specific motifs. Interestingly, HOX and FOX transcription factor families have been shown to have undergone over-expansion in humans⁶. In contrast, conserved CpGs show overrepresentation for TF motifs belonging to families defined by multiple Zinc finger factors, ETS-related and Tal-related factors which are rare among the motifs that are characteristic of human-derived CpGs. In summary, our results indicate a differential contribution of TF families to the transcriptional regulation of conserved and human-derived CpGs in the human genome. We added these results in the revised manuscript. Currently, data on TF binding to methylated and unmethylated CpGs in germlines are lacking in human and non-human primates, thus prohibiting us from investigating whether the binding between methylated and unmethylated CpGs affect evolutionary conservation. However, we will investigate these aspects once such data become available.

Response Figure 3 (also shown as **Supplementary Figure 6** in the revised article). Transcription factor families with differential motif enrichment in conserved CpGs (gray dots) vs. variable-CpGs (blue dots) in (A) core v11 and (B) full HOCOMOCO v11 databases.

4.Human-specific hypo CG-DMRs is a very important finding from this study. Although some examples were discussed, the current manuscript did not provide a clear answer to the question of whether these human specific DMRs were predominantly driven by human-

specific gene overexpression, as shown in the case of *CLUL1*, or local sequence variants (i.e. methylQTL) also have a significant contribution. There may be several ways to address this question - first the authors should analyze the location of these DMRs. if these DMRs are primarily driven by gene overexpression in human, then likely most human specific DMRs are located within gene bodies and enriched at the immediate downstream of TSS. Alternatively, if many of the human specific DMRs are clearly distal regulatory elements, then these DMRs may indicate certain human-specific regulatory program, which could be identified by TF binding motif analysis.

Response: We consider our results of human specific hypo CG-DMR as one of the key findings. Our results indeed demonstrate the regulatory potentials of human specific hypo CG DMRs. Based on our analyses, human specific hypo CG DMRs are involved in regulation by **both distal and proximal mechanisms**. First, we found that human specific hypo CG DMRs are enriched in distal regulatory elements (enhancers). In Response Figure 4, we show that human hypo CG DMRs are enriched in enhancers, particularly for brain enhancers, suggesting that that newly evolved human specific DMRs encode potential for gene regulation. This figure is now included as Supplementary Figure 11 in the revised manuscript.

Response Figure 4 (also shown as **Supplementary Figure 11** in the revised manuscript). Enhancer enrichment at human-specific hypomethylated CG DMRs. 25 chromatin state-model maps based on 6 chromatin mark ChIP-Seq experiments (H3K4me3, H3K4me1, H3K36me3, H3K27me3, H3K9me3 and H3K27ac) were obtained from the Roadmap Epigenomics Project. Each dot represents the enrichment for enhancer-related states (TxReg, TxEnh5', TxEnh3', TxEnhW, EnhA1, EnhA2, EnhW1, EnhW2, and EnhAc) compared to 100 sets of GC-content matched control DMR sets for a given cell-type or

tissue. The original 117 cell-type and tissue-types were grouped into 11 categories shown in the y-axis (total number of cell-type/tissues per group is indicated). $P < 0.01$ for all enrichments.

Second, we also observed that human hypo CG DMRs are enriched in the promoter regions of DEGs, indicating their roles in proximal cis-regulation. In the Response Table 1 below (Supplementary Table 9 in the revised article), we have examined human hypo- and hyper- CG DMRs' associations with up- and down- regulation. Human specific hypo CG DMRs are enriched in genes that are up-regulated in humans in both neurons and oligodendrocytes, while other types of DMRs do not show significant associations. These results support the role of human specific hypo CG methylation on up-regulation.

Response Table 1 (also shown as **Supplementary Table 9** in the revised article). Human hypomethylated CG DMRs are enriched in human-specific up-regulation. Values discussed mainly in the text are shown in bold and italic.

DEG type	DMR-gene type	# of genes ¹	Odds ratio	P-value
Human-UP NeuN+	mCG Human-hyper NeuN+	9037;313;262;5	0.54	9.5E-01
Human-UP NeuN+	mCG Human-hypo NeuN+	9037;313;717;37	1.52	9.0E-03
Human-DOWN NeuN+	mCG Human-hyper NeuN+	9037;181;262;5	0.95	6.1E-01
Human-DOWN NeuN+	mCG Human-hypo NeuN+	9037;181;717;11	0.76	8.6E-01
Human-UP OLIG2+	mCG Human-hyper OLIG2+	9037;354;298;10	0.85	7.4E-01
Human-UP OLIG2+	mCG Human-hypo OLIG2+	9037;354;552;31	1.46	2.7E-02
Human-DOWN OLIG2+	mCG Human-hyper OLIG2+	9037;179;298;3	0.50	9.4E-01
Human-DOWN OLIG2+	mCG Human-hypo OLIG2+	9037;179;552;10	0.91	6.6E-01

¹ # of all orthologous genes expressed in NeuN+; # of DEGs; # of DMR-genes; # of overlap between DEGs with DMR-genes

Third, to directly examine sequence characteristics associated with hypo DMRs, we identified the TF motifs enriched at human-specific hypomethylated DMRs. Specifically, we applied the MEME suite's AME software and two HOCOMOCO v11 databases to compare human-hypomethylated DMRs to chimpanzee-specific hypomethylated DMRs. We found 3 TF motifs significantly associated with human hypomethylated DMRs, including two Forkhead box factors (FOXP1 and FOXP1) and the nuclear factor 1, NFIC. We note that this is not due to the enrichment of variable CpGs, as these DMRs were enriched for both variable and conserved CpGs. Remarkably, FOXP1 is considered a hub gene in human-specific transcriptional networks in the brain and is implicated in several cognitive diseases in humans, including language, intellectual disability and autism⁷.

79% of human-hypomethylated DMRs showed a hit in any of the three TF motifs. A total of 1996 human-specifically hypomethylated DMRs associated with FOXP1 motif, 1906 DMRs with FOXP1 and 462 with NFIC motif. The DMRs with positive hits were highly shared

among TFs, with around 80% shared between FOXP1 and FOXP1, and around 60% of NFIC binding-DMRs also bind the other two TFs. These observations suggest potential co-regulation mechanisms by different transcription factors at human-derived hypomethylated DMRs in the human brain. We hypothesized that the presence of motifs enriched at human-specific hypomethylated DMRs might indicate a stronger biological effect of these DMRs. We found that human-hypomethylated DMRs harboring differentially enriched motifs tend to show larger hypomethylation (Response Figure 5A and 5B). Moreover, the genes associated with these TF-binding DMRs also showed increased gene expression levels compared to chimpanzee brains (Response Figure 5C and 5D).

Response Figure 5 (also shown as **Supplementary Figure 13** in the revised manuscript). (A) Distribution of DNA methylation differences between human and chimpanzee brain

cell-types at human-specific hypomethylated DMRs with and without enriched TF motifs. P-values for Wilcoxon signed-ranked test with alternative = greater. (B) Same as in panel A but separated by the specific TF enriched. (C) Distribution of gene expression differences between human and chimpanzee at human-specific hypomethylated DMRs with and without enriched TF motifs. P-values for Wilcoxon signed-ranked test with alternative = greater. (D) Same as in panel C but separated by the specific TF enriched.

5. How many variants across the three primate species also overlap with human methyl-QTL sites? I went back to an earlier paper by the same group of authors (Mendizabal et al., 2019, Genome Biology). It was somewhat surprising that no methyl-QTL analysis was performed in the earlier work. The question here is whether the DMRs showing variable methylation across primates are also variable across the human population and whether there is a genetic (sequence) underpinning.

Response: We believe that this is a critical question, namely, what are the determinants of DNA methylation variation, and whether the determinants have some similarities within populations as well as between species. While we agree that this is a fundamental and critical question in understanding how DNA methylation and epigenome evolve, and ultimately how regulatory evolution proceeds, the best data set to answer such a question would be a large-scale compilation of DNA methylation variation within human and chimpanzee populations. Unfortunately, even though we have amassed a large number of non-human primate samples, our data might not have sufficient sample size to reliably detect within species variation of DNA methylation. Indeed, in a previous version of our earlier study⁸, we have initially performed an mQTL study. However, after the comments from the reviewers and consultation with the editors, we excluded those results, due to the concern for the small sample size. Rather, our data set is well suited to identify differential DNA methylation between species, which by definition are those that show low within species variation and high between species fixed differences. As shown in Methods, we accounted for factors that can cause within-species variability including gender and age, and identified those that show high species difference after accounting for variation caused by other factors.

Nevertheless, to address the question of potential genetic determinants of DNA methylation between species, we utilized a large-scale mQTL data set generated in Ng et al.⁹. Using 411 samples, they identified 693,696 mQTLs that have significant correlation with DNA methylation of CpGs within 5kb. We asked whether regions surrounding these mQTLs tend to be more evolutionarily divergent than other regions. We compared conservation scores between mQTLs and a control set. Specifically, we selected the strongest SNP for each CpG assuming that each CpG is under the genetic control of a single local SNP and that the rest of the signals nearby are due to linkage disequilibrium. We also removed duplicate SNPs that are associated with more than one methyl loci. We

generated the control set with the same number of loci in the curated mQTLs (n= 42,292) accounting for variation in the minor allele frequencies (MAF). Allele frequency data for common SNPs were downloaded from 1000 Genomes Project (phase 3). Response Figure 6 below shows the mean conservation scores of 200 bp of the focal mQTLs and the control set. We did not see a significant difference between mQTLs and the control set in terms of evolutionary conservation. This observation suggests that sites that modulate within species variation of DNA methylation are not under strong purifying selection.

Response Figure 6. Comparison of mean conservation scores between mQTLs and control CpGs.

Nevertheless, given the strong evidence in the literature that DNA methylation is associated with underlying genetic sequences¹⁰, we further investigated whether sequences that are in the vicinity of mQTLs might have a tendency to affect DNA methylation levels between species. Specifically, we asked whether human mQTLs are more often associated with evolutionary DMRs. We found that among 42,292 mQTLs, 9,988 were closely located with DMRs that show significant methylation differences between humans and chimpanzees, which is a significant excess compared to the MAF-matched control set (odds ratio = 1.75, P-value < 10^{-10} , Fisher's exact test). Since GC contents are known to affect sequence conservation (specifically, GC content is negatively correlated with evolutionary conservation) and regions with higher GC content tend to be more associated with DMRs, we examined the enrichment of mQTLs across differing GC contents and show that these are robust (Response Figure 7, below). Finally, we examined whether CpG sites under mQTL control in humans significantly overlapped with evolutionary DMRs. Out of 55,661 mQTL targeted CpG sites, 2,724 CpGs co-localized with DMRs (a significant enrichment, odds ratio = 2.32, P-value < 10^{-10} , Fisher's exact test), meaning that these sites exhibit variable methylation within human populations and across species. Taken together, these observations suggest that some genomic regions might be more susceptible to genetic changes that affect DNA methylation.

Response Figure 7 (also shown as **Supplementary Figure 9** in the revised manuscript). Genomic windows (200 bps each) containing mQTLs are more often associated with DMRs than genomic windows containing SNPs matched for their minor allele frequency (MAF) (control) of the same size across different GC contents.

We have revised the manuscript to include these results in several sections, as well as in the Supplementary files.

6. When analyzing gene body CH methylation, the authors have preliminary used the raw CH methylation ratio instead of normalizing to global CH methylation level of each species. This would by default leads to more hyper-CH methylated genes in the human. Do the conclusions in Fig. 3 holds if the authors use normalized CH methylation levels (normalized against genome average) for the analysis?

Response: As the reviewer suggested, if overall CH methylation levels show species difference, differential methylation analysis without accounting for this could lead to misleading representations. We examined whether the pattern of DNA methylation between species is different using randomly selected 1 million cytosine positions. Overall, CH methylation distributions are highly similar between humans and chimpanzees, as shown in Response Figure 8.

Response Figure 8. Density plot of DNA methylation levels of randomly selected (1 million) CH sites in humans and chimpanzees. We show two zoomed-in versions to show

that the patterns are consistent. Note that the majority of CH sites are similarly unmethylated in the two species.

The figure above illustrates that the global CH methylation patterns are similar between human and chimpanzee brains. In other words, the proportions of hypermethylated CH sites are very small, even though they are significantly increased in the human brain.

Nevertheless, following the reviewer's suggestion, we performed a quantile normalization across all samples to check whether the result is consistent. Response Figure 9 below shows that gene body CH methylation levels before and after quantile normalization are nearly identical for both species (Response Figure 9A and 9B). Furthermore, gene body CH methylation of humans and chimpanzees show similarly high correlation before and after the quantile normalization (Response Figure 9C and 9D). In addition, CH DMR-genes before and after quantile normalization show highly similar results (100% of the CH-DMR genes, 487 in total, remain significant in the new set after quantile normalization).

Response Figure 9. Correlation coefficient plots displaying the average gene body methylation between before and after quantile normalization. We compared (A) raw and quantile normalized human CH gene body methylation, (B) raw and quantile normalized chimpanzee CH gene body methylation, (C) raw CH gene body methylation of human and

chimpanzee, and (D) quantile normalized CH gene body methylation of human and chimpanzee.

Consequently, we conclude that our results are not affected by background CH methylation. We also note that using raw CH methylation levels make our results comparable to other studies that use similar metrics (e.g., ^{11,12}, others).

7. The results shown in Fig.2 and Fig.3 are somewhat contradictory to one another, which could be due to that raw instead of normalized CH methylation level was used in Fig.3. Presumably if the authors find human specific CG-DMRs overlapping with highly expressed genes such as CLUL1, the gene should also have a lower CH methylation level due to the inverse correlation between CH methylation and gene expression. In any case, it would be good to clarify whether the gene sets whose expression showing a strong correlation with CG and CH methylation are overlapped or not.

Response: The comment by the reviewer motivated us to more clearly present our results indicating **independent, additive roles of CG and CH methylation for regulatory evolution**.

The reviewer is correct that CG and CH methylation of the same genes, for the same genomic regions, are generally correlated, and that gene expression is negatively correlated with both CG and CH methylation (Response Figure 10, new Supplementary Figure 16). However, the correlations vary depending on cell types, cytosine contexts, and whether it is gene body or promoters. Notably, for CG methylation, promoter methylation is the strongest factor for gene expression, and for CH methylation, gene body methylation is the strongest factor. In terms of the correlation between CG and CH methylation, it is higher for gene bodies (correlation coefficients 0.57), it is not as strong between promoter and gene body (correlation coefficients 0.31).

Response Figure 10 (also shown as **Supplementary Figure 16** in the revised manuscript). Correlation coefficient between different methylation contexts and between methylation and gene expression in human NeuN+.

Because of the significant correlations between different factors, we need to use methods to detect independent effects among a set of highly correlated variables (e.g., ¹³). When we perform partial correlation analysis, while promoter CG methylation and gene body CH methylation both remain strongly negatively correlated with gene expression, gene body CG methylation changes its directionality (Response Table 2). Multiple linear regression analyses also provide consistent results (Response Table 3). These are consistent with previous studies ¹⁴⁻¹⁶ and may indicate consequential impact of transcription on gene body CG methylation, and/or its roles on alternative transcription, which are beyond the scope/focus of our current paper.

Response Table 2 (Supplementary Table 13). Partial correlation analysis explaining correlation coefficients between methylation and expression while accounting for effects from other methylation contexts.

Predictors	Gene expression	
	Normal ¹ (P-value)	Partial ² (P-value)
mCG promoter	-0.36 ($P < 10^{-10}$)	-0.27 ($P < 10^{-10}$)
mCG gene body	-0.22 ($P < 10^{-10}$)	0.18 ($P < 10^{-10}$)
mCH gene body	-0.53 ($P < 10^{-10}$)	-0.48 ($P < 10^{-10}$)

¹ordinary correlation coefficient (Spearman) ²partial correlation coefficient (Spearman)

On the other hand, promoter CG methylation and gene body CH methylation have been proposed to play causative roles in regulation of gene expression^{11,17-19}, and show strong and independent effects on gene expression difference between species (Response Table 3). Therefore, these analyses support independent, additive significance of 1) human-specific CG hypomethylation on cis-regulatory regions (including enhancers and promoters) promoting up-regulation of specific genes, and 2) human-specific CH hypermethylation on gene bodies facilitating down-regulation of specific genes.

Response Table 3 (Supplementary Table 14). Multiple linear regression models explaining variation of gene expression levels of humans and human-chimpanzee difference.

Predictors	Estimate of β	t-value	Significance	
Human data alone				
Intercept	3.05	67.15	$< 10^{-15}$	
Promoter mCG	-1.46	-33.28	$< 10^{-15}$	
Gene body mCG	2.20	29.38	$< 10^{-15}$	
Gene body mCH	-44.14	-65.74	$< 10^{-5}$	
Adj-R²				0.39
Human-chimpanzee difference				
Intercept	0.005	1.83	0.06	
Promoter mCG difference	-0.17	-3.19	0.001	
Gene body mCG difference	0.21	1.91	0.06	
Gene body mCH difference	-11.72	-19.32	$< 10^{-10}$	
Adj-R²				0.14

To illustrate and test these effects more directly, below we show the relationship between CH gene body methylation difference (X-axis) and promoter CG methylation difference (Y-axis) between human and chimpanzee data (Response Figure 11). For the X-axis (gene body CH methylation), we observe greater density for positive X-values, indicating human hypermethylation. For the Y-axis (promoter CG methylation), we observe greater density for negative Y-values, indicating human hypomethylation. When we mark human up-regulated genes, these are significantly enriched in the promoter-hypomethylated genes (3rd and 4th quadrants, $P = 0.0017$, Fisher's exact test). In contrast, human down-regulated genes are significantly enriched in the gene body CH hyper-methylated genes (1st and 4th quadrants, $P = 2.5 \times 10^{-10}$, Fisher's exact test).

Response Figure 11 (Supplementary Figure 17). Relationship between CH gene body methylation (relative difference between species, X-axis) and CG promoter methylation (relative difference between species, Y-axis) in different gene types.

In addition, we have previously identified key modules that show human-specific changes in neuron co-expression networks²⁰. We observe that two human up-regulated modules (NM16 and NM21) are significantly enriched with genes harboring CG hypo DMRs in their promoters. On the other hand, all three human-downregulated modules (NM3, NM14, NM20) are enriched in human CH-gene body hypermethylated genes (Response Table 4).

Response Table 4 (Supplementary Table 15). Significance levels of human-specific DMR-genes for human-specific WGCNA modules. Values discussed mainly in the text are shown in bold and italic.

Human-specific WGCNA modules	module	# of genes	mCG DMR gene				mCH DMR gene			
			hyper		hypo		hyper		hypo	
			P-value	OR	P-value	OR	P-value	OR	P-value	OR
Up regulation	NM2	847	9.9E-01	0.51	5.0E-01	1.02	8.9E-01	0.78	2.6E-02	4.24
	NM16	294	4.6E-01	1.11	3.4E-03	2.03	9.3E-01	0.56	1.0E+00	0.00
	NM21	313	6.9E-01	0.87	1.4E-02	1.80	9.5E-01	0.53	2.6E-02	7.64

Down	NM3	330	5.7E-01	0.99	7.9E-01	0.80	2.7E-03	2.12	1.0E+00	0.00
regulation	NM14	148	5.5E-02	2.20	8.3E-01	0.67	2.4E-04	3.34	1.0E+00	0.00
	NM20	887	2.0E-01	1.23	6.0E-01	0.97	2.3E-02	1.44	1.0E+00	0.00

Therefore, these observations indicate independent, and additive roles of CG hypomethylation of cis-regulatory regions and CH hypermethylation of gene bodies during human brain evolution. We have revised the manuscript to more clearly emphasize these points, and included all these figures and tables as additional supplementary material.

8. *The partitioned heritability analysis is standard for this type of study and is also informative. Can the authors provide an example and a list of the overlaps between schizophrenia credible SNPs (from finemapping) and DMRs?*

Response: Following the reviewer's suggestion, in the revised manuscript we provide a list of human-specific neuron hypomethylated DMRs and a list of conserved neuron-hypomethylated DMRs (both categories with significant enrichment in LDSC for SZ heritability) containing credible schizophrenia signal. Specifically, under a P-value cutoff of 10^{-5} reported in the original GWAS, we identify a total of 53 and 432 SNPs at 26 human-specific and 199 conserved neuron hypomethylated DMRs, respectively. In addition to several DMRs at the MHC region in chromosome 6 (not considered in LDSC), we also report evolutionary DMRs at other bona-fide schizophrenia genes such as at *CACNA1C*. We reference this table (new Supplementary Table 19) in the revised text and methods.

Supplementary Table 19. List of SZ GWAS SNPs (PMID: 25056061) with P-value below 10^{-5} located at human-specific neuron hypomethylated DMRs (53 SNPs) and conserved neuron hypomethylated DMRs (432 SNPs).

Reviewer #3 (Remarks to the Author):

In "Evolution of DNA Methylation in the Human Brain", the authors generate DNA methylation data from neurons and oligodendrocytes in the prefrontal cortex of human, chimp, and macaque. They identify conserved and species-specific sites of mCG and mCH and increased mCH and decreased mCG in the human lineage. They increase the interpretability of their finding by integrating their data with several publicly available datasets. They show that these methylation changes may act to change gene expression at specific developmental timepoints and in a specific classes of cortical neurons. Surprisingly, they find that genomic loci associated with many brain disorders and traits are enriched in conserved mCG sites and only schizophrenia loci were enriched in human-specific mCG sites. This paper provides a useful dataset to the scientific community interested in comparative genomics and human brain disease. The results would be strengthened by adding a power analysis to show that

differences in the enrichment of brain traits for different sets of mCH and mCG sites are not driven by differences in the number of risk loci and/or sites.

Response: We appreciate the constructive comments by the reviewer. Regarding the power analyses on disease enrichment at the different DMR categories, these were included in the previous version but not clearly referenced in the main text (additional comments are following specific comments by this reviewer on Figure 4A below, in page 19-20 in this response letter).

Specific comments:

- *SI Fig. 5 – Please move these data to main Fig. 1F and test whether there is a significant difference between human and chimp. If not, this suggests that evolutionary rates are similar along the two lineages.*

Response: Following this suggestion, we have moved the supplementary figure to the main figure (Figure 1E in the revised manuscript). Also, we separated the DMRs into the two cell types. By doing so, the larger DNA methylation difference between cell types in evolutionarily old DMRs (i.e. DMRs conserved in all three species) compared to recently evolved DMRs in humans or chimpanzees is more clearly perceived.

- *Discussion – Please comment on the possibility that some species-specific differences in the NeuN+ population may be driven by evolutionary changes in proportions of different cell types in DLPFC. For example, a potentially greater expansion of supragranular neurons in human vs. chimp.*

Response: Please see our response to the first comment of the reviewer 1. We have revised our manuscript to provide a more nuanced interpretation including the alternative possibility of different cell proportions of human and chimpanzee DLPFC.

- *Fig. 3d/SI Fig. 13 – Macaque CH DMR genes have lower expression in macaque than human throughout development. Please comment on this difference during fetal development vs. human CH DMR genes. What are the results of a similar analysis using human CG DMR genes? Do you see lower expression of human than macaque during prenatal development, suggesting that this CG methylation persists through development?*

Response: Since we do not have an outgroup, we cannot classify if DMRs in macaques are derived or ancestral. Therefore, genes referred to as ‘macaque CH DMR genes’ in the previous version of the manuscript could have arisen due to conserved ancestral hypermethylation or derived methylation in the macaque lineage, and should not have been indicated as macaque CH DMR genes. We apologize for this oversight and thank the

reviewer for drawing our attention to this important point. Due to the uncertainty of the evolutionary directionality, our interpretation is limited. We thus removed the mention of this group of genes in the revised Supplementary Figure 18. On the other hand, comparison of human and chimpanzee CH DMR genes remains. As a complementary analysis to the CH DMR comparison between humans and chimpanzees, we also performed comparison of gene expression of CG DMR genes and found that there was no significant difference between human and macaque throughout development even though prenatal 13 PCW shows slight human up-regulation compared to macaque (Response Figure 12).

Response Figure 12 (Supplementary Figure 18). Gene expression in human and macaque for CG DMR genes (hypo and hyper) over developmental time points. Macaque samples were age-matched to human developmental time points (PCW: Post conception week; M: Month; Yr: Year).

• Fig. 4A – Could the lack of significant enrichment for diseases in species-specific mCH/mCG be due to a lack of power? If you downsample conserved NeuN+ hypo mCG sites to be the same number as human-specific sites, do you still find significant enrichment for brain diseases?

Response: We would like to emphasize that we do observe significant enrichment for schizophrenia in human-specific NeuN CG hypomethylated regions (Figure 4A). We appreciate the reviewer's comment, as we were also concerned about decreased power due to a lower number and shorter length of human-specific DMR categories. This motivated us to perform down-sampling analyses that were reported in the previous version (previously as Supplementary Figure 16, now Supplementary Figure 21 in the revised manuscript). This analysis showed that after down-sampling conserved CG DMRs to the number and length of human-specific CG DMR, the former do still show stronger enrichment for schizophrenia heritability. Therefore, we conclude that differences in the DMR length cannot explain the more diluted signal on human-specific CG DMRs. Regarding CH, there is no such a concern on power since both human-specific and conserved NeuN hyper DMRs show similar significant depletion.

We apologize these analyses were buried in methods and supplementary material in the previous version. In the revised version we mention these analyses in the main text:

“Moreover, human-specific neuron-hypo CG DMRs exhibited significant enrichment for schizophrenia heritability (Fig. 4a, 4b even though the degree of enrichment is lower than that for the conserved DMRs as suggested by down-sampling analyses (Supplementary Fig. 21). “

It is surprising that a disease that affects only humans has risk loci found in conserved sites. How do you interpret this?

Response: Our observation of a large contribution of conserved brain epigenetic marks to schizophrenia is in line with other recent studies. For example, Hujoel et al. ²¹ reported the largest contributions of ancient enhancers and promoters to complex diseases compared to more recently evolved regulatory regions. In general, genomic regions under strong and ancestral purifying selection (thus remain conserved) are enriched for disease genes and heritability ²¹⁻²³. The fact that schizophrenia shows a similar pattern to other complex diseases suggests that even though the phenotypes of schizophrenia is highly specific to humans, the molecular and developmental mechanisms of this disease are likely to have deep phylogenetic roots. For example, increasing number of studies including ours has linked developmental dysregulation to schizophrenia pathology ^{8,12,24}. We also note that many neurodevelopmental and neuropsychiatric disorders are known to share common variants according to GWAS studies. In addition, we found significant impact of newly evolved, human brain-specific differential DNA methylation to schizophrenia heritability. This observation indicates that while the majority of molecular pathways have been shaped by purifying selection, recent, human specific evolutionary changes contribute to schizophrenia pathology. Thus, our results contribute to understanding the relevance of

conserved and derived regulatory mechanisms to the architecture of a complex disease. We have updated the discussion section in the revised manuscript to provide a more nuanced interpretation.

Likewise, how does the number of risk loci compare across diseases?

SCZ GWAS have more subjects than many other traits/diseases, and the disease specificity you see in human-specific sites may be driven by increased power to detect enrichment.

Response: We would like to point out that in the LDSC analysis all genome-wide genotyped SNPs are analyzed, without filtering based on significance in the original GWAS. Therefore, the number of SNPs analyzed does not vary depending on the significant GWAS loci. It is certainly possible that some diseases show lack of significance in LDSC analyses due to low sample size in the original GWAS study. However, there is no trend in our data suggesting such a bias. Below (Response Figure 13) we directly plotted the relationships (A) between the number of GWAS significant loci and the FDR-corrected significance in the LDSC using NeuN hypo DMRs, as well as (B) the relationship between the sample size and the FDR-corrected significance in the LDSC. For example, low sample-sized GWAS like Alzheimer's and Anorexia show significance in conserved hypomethylated CG DMRs, whereas other well-empowered GWAS such as height (larger sample sizes than schizophrenia) do not, suggesting a coordinated signature exclusively in brain-related diseases (Response Figure 13B). In any case, due to this uncertainty, please note that we do not draw conclusions on the relative impact of epigenomic divergence across different brain-diseases, but rather aim to compare the contribution of different DMR categories to well-powered brain diseases such as schizophrenia.

Response Figure 13. Relationship between GWAS significant loci (A) and sample sizes (B) with the LDSC analyses for conserved NeuN Hypo DMRs in our data.

• *Fig. 4B – Does the depletion of SCZ SNPs in human-specific mCH hyper sites support a role for inhibitory neurons in the disease given your finding that these sites are associated with inhibitory marker genes?*

Response: Our results point out a significant depletion of SZ risk at mCH hyper sites. Since we found these signals were enriched at inhibitory markers, we agree that this could point to a lack of a role of inhibitory genes in schizophrenia risk. Indeed, a previous study using human expression data and GWAS variants proposed a greater role of excitatory neurons over inhibitory neurons for schizophrenia risk²⁵. Analyses of open chromatin of different cell populations also suggest a greater role of excitatory neurons²⁶. We have added this point in the revised manuscript.

• *Please comment on the extent to which differences in genome annotation quality could affect your quantification of DNA methylation and conclusions about excess/depletion of mCG/mCH hyper/hypomethylation. Previous work has suggested that human neurons expression more genes than chimpanzee neurons, but a recently improved chimpanzee genome annotation has resulted in more accurate transcript quantification and suggests that overall transcript levels are quite comparable (see Pollen et al. 2019 Cell).*

Response: We have used well-annotated genome assemblies for both human and non-human primates. Total numbers of CG positions we examined for human, chimpanzee, and macaque genome assemblies are 26.7 million, 27 million, and 27.7 million, respectively, reflecting that the quality of genome annotation is similar across species.

Following the reviewer's comment, we examined the extent to which genome annotation qualities have an impact on the pattern of human-specific CG and CH methylation changes. We re-ran the differential methylation analysis using a previous version of the human assembly (hg17) to see if the patterns are consistent. Of note, this version of human genome assembly lacks 5.5 million cytosine positions which are present in the hg19 assembly.

Human-specific DMRs show the consistent excess of neuron-hypomethylated CG DMR and neuron-hypermethylated CH DMR (shown in Response Table 5), and more than 95% of DMRs remain. Taken together, these observations suggest that the human-specific methylation patterns reported in the study are robust to variable genome annotation qualities.

Response Table 5. Number of species-specific DMR using different genome versions.

	# of DMR (hg19)	# of DMR (hg17)
CG DMR		
Human-specific neuron-hyper	1808	1807
Human-specific neuron-hypo	6363	6358
Chimpanzee-specific neuron-hyper	2804	2801
Chimpanzee-specific neuron-hypo	3499	3495
Human-specific oligodendrocyte-hyper	3254	3250
Human-specific oligodendrocyte-hypo	5553	5549
Chimpanzee-specific oligodendrocyte-hyper	3105	3103
Chimpanzee-specific oligodendrocyte-hypo	5178	5172
CH DMR		
Human-specific hyper	7391	7373
Human-specific hypo	1240	1239
Chimpanzee-specific hyper	4566	4558
Human-specific hypo	1378	1375

Literature Cited in Response

- 1 Bird, A. DNA methylation and the frequency of CpG in animal DNA. *Nucleic Acids Res.* **8**, 1499-1504 (1980).
- 2 Mendizabal, I. & Yi, S. V. Whole-genome bisulfite sequencing maps from multiple human tissues reveal novel CpG islands associated with tissue-specific regulation. *Human Molecular Genetics* **25**, 69-82, doi:10.1093/hmg/ddv449 (2016).
- 3 Kim, S.-H., Elango, N., Warden, C. W., Vigoda, E. & Yi, S. Heterogeneous genomic molecular clocks in primates. *PLoS Genetics* **2**, e163 (2006).
- 4 Bailey, T. L. *et al.* MEME Suite: tools for motif discovery and searching. *Nucleic Acids Research* **37**, W202-W208 (2009).
- 5 Kulakovskiy, I. V. *et al.* HOCOMOCO: expansion and enhancement of the collection of transcription factor binding sites models. *Nucleic Acids Research* **44**, D116-D125, doi:10.1093/nar/gkv1249 (2016).
- 6 Rosanova, A., Colliva, A., Osella, M. & Caselle, M. Modelling the evolution of transcription factor binding preferences in complex eukaryotes. *Scientific reports* **7**, 7596-7596, doi:10.1038/s41598-017-07761-0 (2017).
- 7 Konopka, G. *et al.* Human-specific transcriptional networks in the brain. *Neuron* **75**, 601-617 (2012).
- 8 Mendizabal, I. *et al.* Cell type-specific epigenetic links to schizophrenia risk in the brain. *Genome Biology* **20**, 135, doi:10.1186/s13059-019-1747-7 (2019).
- 9 Ng, B. *et al.* An xQTL map integrates the genetic architecture of the human brain's transcriptome and epigenome. *Nat Neurosci* **20**, 1418-1426, doi:10.1038/nn.4632 (2017).
- 10 Yi, S. V. Insights into Epigenome Evolution from Animal and Plant Methylomes. *Genome Biol Evol* **9**, 3189-3201, doi:10.1093/gbe/evx203 (2017).
- 11 Lister, R. *et al.* Global Epigenomic Reconfiguration During Mammalian Brain Development. *Science* **341**, 1237905 (2013).

- 12 Price, A. J. *et al.* Divergent neuronal DNA methylation patterns across human cortical development reveal critical periods and a unique role of CpH methylation. *Genome Biology* **20**, 196, doi:10.1186/s13059-019-1805-1 (2019).
- 13 Kim, S.-H. & Yi, S. Understanding relationship between sequence and functional evolution in yeast proteins. *Genetica* **131**, 151-156 (2007).
- 14 Huh, I., Zeng, J., Park, T. & Yi, S. DNA methylation and transcriptional noise. *Epigenetics & Chromatin* **6**, 9 (2013).
- 15 Jjingo, D., Conley, A. B., Yi, S. V., Lunyak, V. V. & Jordan, I. K. On the presence and role of human gene-body DNA methylation. *Oncotarget* **3**, 462-474 (2012).
- 16 Spainhour, J. C. G., Lim, H. S., Yi, S. V. & Qiu, P. Correlation Patterns Between DNA Methylation and Gene Expression in The Cancer Genome Atlas. *Cancer Informatics* **18**, 1176935119828776, doi:10.1177/1176935119828776 (2019).
- 17 Rizzardi, L. F. *et al.* Neuronal brain-region-specific DNA methylation and chromatin accessibility are associated with neuropsychiatric trait heritability. *Nat Neurosci* **22**, 307-316, doi:10.1038/s41593-018-0297-8 (2019).
- 18 Stroud, H. *et al.* Early-Life Gene Expression in Neurons Modulates Lasting Epigenetic States. *Cell* **171**, 1151-1164.e1116, doi:10.1016/j.cell.2017.09.047 (2017).
- 19 Schübeler, D. Function and information content of DNA methylation. *Nature* **517**, 321-326, doi:10.1038/nature14192 (2015).
- 20 Berto, S. *et al.* Accelerated evolution of oligodendrocytes in the human brain. *Proceedings of the National Academy of Sciences* **116**, 24334, doi:10.1073/pnas.1907982116 (2019).
- 21 Hujoel, M. L. A., Gazal, S., Hormozdiari, F., van de Geijn, B. & Price, A. L. Disease Heritability Enrichment of Regulatory Elements Is Concentrated in Elements with Ancient Sequence Age and Conserved Function across Species. *The American Journal of Human Genetics* **104**, 611-624 (2019).
- 22 Domazet-Lošo, T. & Tautz, D. An Ancient Evolutionary Origin of Genes Associated with Human Genetic Diseases. *Molecular Biology and Evolution* **25**, 2699-2707, doi:10.1093/molbev/msn214 (2008).
- 23 Finucane, H. K. *et al.* Partitioning heritability by functional annotation using genome-wide association summary statistics. *Nature Genetics* **47**, 1228-1235, doi:10.1038/ng.3404 (2015).
- 24 Jaffe, A. E. *et al.* Developmental and genetic regulation of the human cortex transcriptome illuminate schizophrenia pathogenesis. *Nat Neurosci* **21**, 1117-1125, doi:10.1038/s41593-018-0197-y (2018).
- 25 Finucane, H. K. *et al.* Heritability enrichment of specifically expressed genes identifies disease-relevant tissues and cell types. *Nature genetics* **50**, 621-629, doi:10.1038/s41588-018-0081-4 (2018).
- 26 Hauberg, M. E. *et al.* Common schizophrenia risk variants are enriched in open chromatin regions of human glutamatergic neurons. *Nature communications* **11**, 5581-5581, doi:10.1038/s41467-020-19319-2 (2020).

Reviewers' Comments:

Reviewer #2:

Remarks to the Author:

The authors have done a great job responding to my questions and made some additional and very interesting findings. I have no additional comments.

Reviewer #3:

Remarks to the Author:

All of my comments have been thoughtfully addressed. I support publication.

Trygve Bakken